# Bayesian inference for biophysical neuron models enables stimulus optimization for retinal neuroprosthetics

Jonathan Oesterle[1], Christian Behrens[1], Cornelius Schröder[1], Thoralf Hermann[2], Thomas Euler[1,3,4], Katrin Franke[1,4], Robert G Smith[5], Günther Zeck[2], Philipp Berens[1,3,4,6]*

[1]Institute for Ophthalmic Research, University of Tübingen, Tübingen, Germany; [2]Naturwissenschaftliches und Medizinisches Institut an der Universität Tübingen, Reutlingen, Germany; [3]Center for Integrative Neuroscience, University of Tübingen, Tübingen, Germany; [4]Bernstein Center for Computational Neuroscience, University of Tübingen, Tübingen, Germany; [5]Department of Neuroscience, University of Pennsylvania, Philadelphia, United States; [6]Institute for Bioinformatics and Medical Informatics, University of Tübingen, Tübingen, Germany

**Abstract** While multicompartment models have long been used to study the biophysics of neurons, it is still challenging to infer the parameters of such models from data including uncertainty estimates. Here, we performed Bayesian inference for the parameters of detailed neuron models of a photoreceptor and an OFF- and an ON-cone bipolar cell from the mouse retina based on two-photon imaging data. We obtained multivariate posterior distributions specifying plausible parameter ranges consistent with the data and allowing to identify parameters poorly constrained by the data. To demonstrate the potential of such mechanistic data-driven neuron models, we created a simulation environment for external electrical stimulation of the retina and optimized stimulus waveforms to target OFF- and ON-cone bipolar cells, a current major problem of retinal neuroprosthetics.

*For correspondence:
philipp.berens@uni-tuebingen.de

## Introduction

Mechanistic models have been extensively used to study the biophysics underlying information processing in single neurons and small networks in great detail (*Gerstner and Kistler, 2002*; *Koch, 2004*). In contrast to phenomenological models used for neural system identification, such models try to preserve certain physical properties of the studied system to facilitate interpretation and a causal understanding. For example, biophysical models can incorporate the detailed anatomy of a neuron (*Golding et al., 2001*; *Poirazi et al., 2003*; *Hay et al., 2011*), its ion channel types (*Hodgkin and Huxley, 1952*; *Fohlmeister and Miller, 1997*) and the distributions of these channels (*Rattay et al., 2017*) as well as synaptic connections to other cells (*O'Leary et al., 2014*). For all these properties, the degree of realism can be adjusted as needed. While increased realism may enable models to capture the highly non-linear dynamics of neural activity more effectively, it usually also increases the number of model parameters. While the classical Hodgkin-Huxley model with one compartment has already 10 free parameters (*Hodgkin and Huxley, 1952*), detailed multicompartment models of neurons can have dozens or even hundreds of parameters (*Taylor et al., 2009*; *Hay et al., 2011*).

Constraining many of these model parameters such as channel densities requires highly specialized and technically challenging experiments, and hence it is usually not viable to measure every single parameter for a neuron model of a specific neuron type. Rather, parameters for mechanistic

simulations are often aggregated over different neuron types and even across species. Even though this may be justified in specific cases, it likely limits our ability to identify mechanistic models of individual cell types. Alternatively, parameter search methods have been proposed to identify the parameters of mechanistic neuron models from standardized patch-clamp protocols based on exhaustive grid-searches (*Goldman et al., 2001*; *Prinz et al., 2003*; *Stringer et al., 2016*) or evolutionary algorithms (*Gerken et al., 2006*; *Keren et al., 2005*; *Achard and De Schutter, 2006*; *Rossant et al., 2011*). Such methods are often inefficient and identify only a single point estimate consistent with the data (for discussion, see *Gonçalves et al., 2020*).

Here, we built on recent advances in Bayesian simulation-based inference to fit multicompartment models of neurons with realistic anatomy in the mouse retina. We used a framework called Sequential Neural Posterior Estimation (SNPE) (*Lueckmann et al., 2017*; *Gonçalves et al., 2020*) to identify model parameters based on high-throughput two-photon measurements of these neurons' responses to light stimuli. SNPE is a Bayesian simulation-based inference algorithm that allows parameter estimation for simulator models for which the likelihood cannot be evaluated easily. The algorithm estimates the distribution of model parameters consistent with specified target data by evaluating the model for different sets of parameters and comparing the model output to the target data. To this end, parameters are drawn from a prior distribution, which is an initial guess about which parameters are likely to produce the desired model output. For example, the choice of prior distribution can be informed by the literature, without constraining the model to specific values. The model output for the sampled parameter sets can then be used to refine the distribution over plausible parameters given the data. This updated distribution, containing information from both the prior and the observed simulations, is known as the posterior. For high-dimensional parameter spaces, many samples are necessary to obtain an informative posterior estimate. Therefore, to make efficient use of simulation time, SNPE iteratively updates its sampling distribution, such that only in the first round samples are drawn from the prior, while in subsequent rounds samples are drawn from intermediate posteriors. This procedure increases the fraction of samples leading to simulations close to the target data. Since this approach for parameter estimation not only returns a point-estimate but also a posterior distribution over parameters consistent with the data, it allows one to straightforwardly determine how well the parameters are constrained. While the method has been used previously to fit simple neuron models (*Lueckmann et al., 2017*; *Gonçalves et al., 2020*), it has so far not been applied to models as complex and realistic as the ones presented here.

We estimated the posterior parameter distribution of multicompartment models of three retinal neurons, a cone photoreceptor (cone), an OFF- and an ON-bipolar cell (BC). The structure of the BC models was based on high-resolution electron microscopy reconstructions (*Helmstaedter et al., 2013*) and in six independently parameterized regions. We performed parameter inference based on the responses of these neurons to standard light stimuli measured with two-photon imaging of glutamate release using iGluSnFR as an indicator (*Franke et al., 2017*). Our analysis shows that many of the model parameters can be constrained well, yielding simulation results consistent with the observed data. After validating our model, we show that the inferred models and the inference algorithm can be used to efficiently guide the design of electrical stimuli for retinal neuroprosthetics to selectively activate OFF- or ON-BCs. This is an important step toward solving a long-standing question in the quest to provide efficient neuroprosthetic devices for the blind.

## Materials and methods

### Key resources table

| Reagent type (species) or resource | Designation | Source or reference | Identifiers | Additional information |
|---|---|---|---|---|
| Genetic reagent (mouse) | B6;129S6-Chat$^{tm2(cre)LowlJ}$ | Jackson laboratory JAX 006410 | RRID:IMSR_JAX:006410 | |
| Genetic reagent (mouse) | Gt(ROSA)26Sor $^{tm9(CAG-tdTomato)Hze}$ | Jackson laboratory JAX 007905 | RRID:IMSR_JAX:007905 | |
| Strain (mouse, female) | B6.CXB1-Pde6b$^{rd10}$ | Jackson laboratory JAX 004297 | RRID:IMSR_JAX:004297 | |

*Continued on next page*

*Continued*

| Reagent type (species) or resource | Designation | Source or reference | Identifiers | Additional information |
|---|---|---|---|---|
| Strain (Adeno-associated virus) | AAV2.7m8.hSyn.iGluSnFR | Virus facility, Institute de la Vision, Paris | | |
| Software, algorithm | NeuronC | https://retina.anatomy.upenn. edu/rob/neuronc.html | RRID:SCR_014148 | Version 6.3.14 |
| Software, algorithm | SNPE | https://github.com/mackelab/delfi | | See Inference algorithm |
| Software, algorithm | COMSOL Multiphysics | COMSOL Multiphysics | RRID:SCR_014767 | |

## Biophysical neuron models

We created detailed models of three retinal cell types: a cone, an ON- (*Figure 1A,Bi*) and an OFF-BC (*Figure 1Bii*). From the different OFF- and ON-BC types, we chose to model the types 3a and 5o, respectively, because those were the retinal cone bipolar cell (CBC) types in mice for which we could gather most information. To model the light response, the OFF-BC model received input from five and the ON-BC from three cones (*Behrens et al., 2016*). Every cone made two synaptic connection with each BC. The postsynaptic conductances were set to 0.25 nS per connection.

### Multicompartment models

We used NeuronC (*Smith, 1992*) to implement multicompartment models of these neurons. A multicompartment model subdivides a neuron into a finite number of compartments. Every compartment is modeled as an electrical circuit, has a position in space, a spatial shape and is connected to at least one neighboring compartment (*Figure 1C*). The voltage in a compartment $n$, connected to the compartments $n-1$ and $n+1$ is described by:

$$\frac{\delta}{\delta t} V_m^n = \frac{1}{c_m^n} \left( \frac{V_m^{n+1} - V_m^n}{r_i^{n+1} + r_i^n} + \frac{V_m^{n-1} - V_m^n}{r_i^{n-1} + r_i^n} + \frac{V_r^n + V_{ex}^n - V_m^n}{r_m^n} + \sum_e \frac{V_e^n + V_{ex}^n - V_m^n}{r_e^n(\ldots)} \right) + \frac{\delta}{\delta t} V_{ex}^n. \tag{1}$$

Here, compartments are either modeled as cylinders or spheres. The membrane capacitance $c_m^n$, membrane resistance $r_m^n$ and axial resistance $r_i^n$ of a compartment $n$ are assumed to be dependent on the compartment surface area $A_m^n$ and/or the compartment length $l_c^n$:

$$r_m^n = R_m / A_m^n, \qquad c_m^n = C_m A_m^n, \qquad r_i^n = R_i l_c^n / A_m^n. \tag{2}$$

We assumed the specific membrane resistance $R_m$, the specific membrane capacitance $C_m$, and the axial resistivity $R_i$ to be constant over all compartments within a cell model. We used $R_i = 132\,\Omega\,\text{cm}$ for all cell types and informed our priors for $C_m$ and $R_m$, which we estimated for every cell type individually, based on estimates for rod bipolar cells of rats (*Oltedal et al., 2009*). Parameters of NeuronC and the used values are summarized in *Appendix 1—table 3*.

### Anatomy

We used a simplified cone morphology consisting of four compartments: one cone-shaped compartment for the outer segment, one spherical compartment for the combination of inner segment and soma, one cylindrical compartment for the axon, and another spherical one for the axonal terminals (*Figure 1*). The light collecting area in the outer segment was set to 0.2 $\mu m^2$ (*Nikonov et al., 2006*). The diameter of the soma $d_S^c$, the axon $d_A^c$ and axonal terminals $d_{AT}^c$, the length of the axon $l_A^c$ and the length of the outer segment $l_{OS}^c$ were based on electron microscopy data (*Carter-Dawson and LaVail, 1979*):

$$d_S^c = 5.13\,\mu\text{m}, \quad d_A^c = 1.3\,\mu\text{m}, \quad d_{AT}^c = 6\,\mu\text{m}, \quad l_A^c = 15\,\mu\text{m}, \quad l_{OS}^c = 14.4\,\mu\text{m}. \tag{3}$$

The BC morphologies in this study were based on serial block-face electron microscopy data of mouse bipolar cells (*Helmstaedter et al., 2013*). We extracted the raw voxel-based morphologies from the segmentation of the EM dataset and transformed them into a skeleton plus diameter

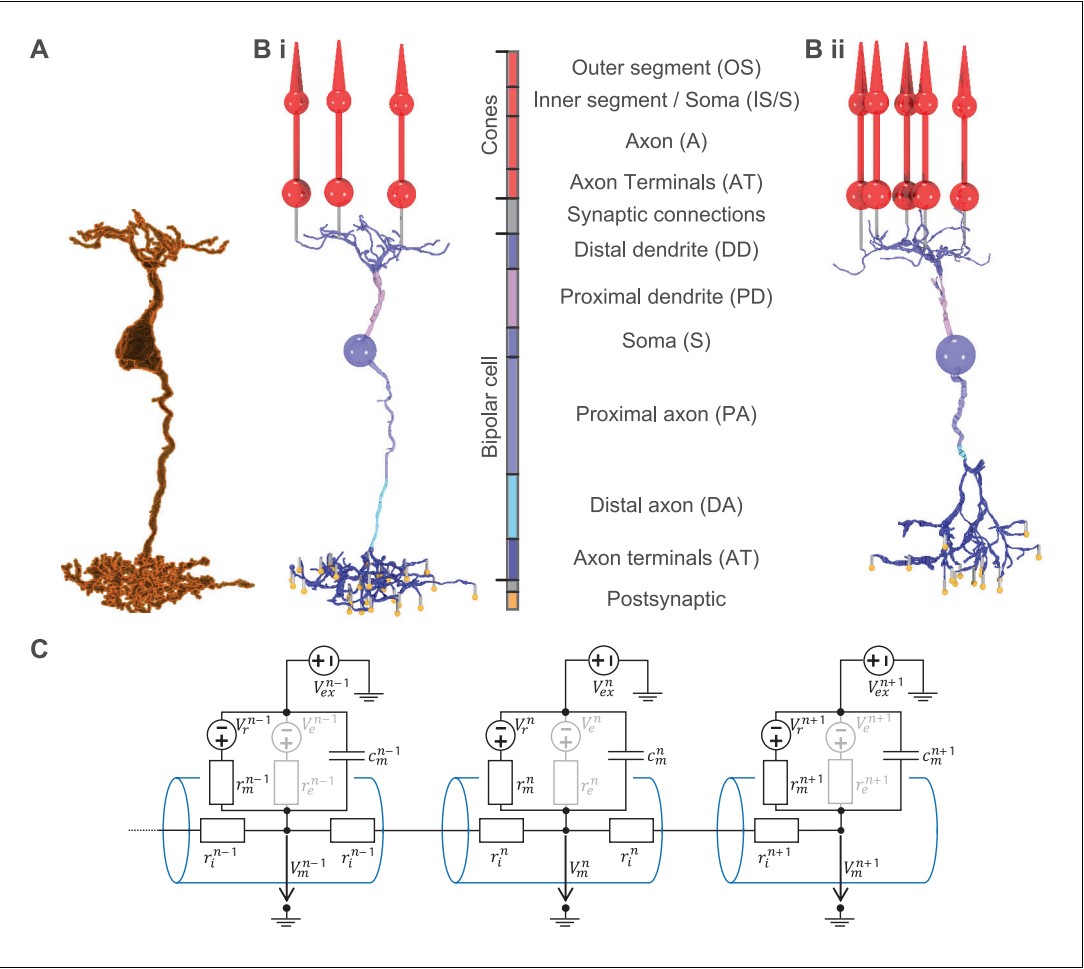

**Figure 1.** From serial block-face electron microscopy (EM) data of retinal BCs to multicompartment models. (**A**) Raw morphology extracted from EM data of an ON-BC of type 5o. (**Bi**) Processed morphology connected to three presynaptic cones (red) and several postsynaptic dummy compartments that are generated to create the synapses in the model (yellow). The cone and BC morphologies are divided into color-coded regions with a legend shown on the right. (**Bii**) Same as (**Bi**) but for an OFF-BC of type 3. (**C**) Three cylindrical compartments of a multicompartment model. Every compartment (blue) $n$ consists of a membrane capacitance $c_m^n$, a membrane resistance $r_m^n$, a leak conductance voltage source $V_r^n$, an extracellular voltage source $V_{ex}^n$ and at least one axial resistor $r_i^n$ that is connected to a neighboring compartment. $V_{ex}^n$ is only used to simulate electrical stimulation and is otherwise replaced by a shortcut. Compartments may have one or more further voltage- or ligand-dependent resistances $r_e^n$ with respective voltage sources $r_e^n$ to simulate ion channels (indicated in gray).

representation using Vaa3D-Neuron2 auto tracing (*Xiao and Peng, 2013*). These where then manually refined using Neuromantic (*Myatt et al., 2012*) to correct errors originating from small segmentation errors (*Figure 1*).

The ON-BC morphology we chose was classified as type 5o, equal to the functional type of the model. For the OFF-BC, we decided for a morphology classified as type 3b, although we functionally modeled a type 3a cell, because the chosen reconstructed morphology was of higher quality than all available type 3a reconstructions and because type 3a and 3b BCs have very similar morphologies. Additionally, type 3a and 3b mostly differ in the average axonal field size (*Wässle et al., 2009*; *Helmstaedter et al., 2013*), with that of type 3a being larger than that of type 3b. The selected morphology had the largest axonal field among all cells classified as 3b in the dataset, well within the range of type 3a cells.

Because the computational time scales approximately linear with the number of BC compartments, using the full number of compartments of the EM reconstructions (>1000) during parameter

inference was computationally infeasible. Therefore, we utilized the compartment condensation algorithm of NeuronC, which iteratively reduces the number of compartments while preserving biophysical properties (*Smith, 1992*). To be able to draw a sufficient number of samples, we reduced the number of compartments during parameter inference to 22 and 19 for the OFF- and ON-BC respectively (requiring approximately 3 min per simulation for a 31 s light stimulus). To simulate the electrical stimulation, more compartments are necessary to capture the effect of the electrical field on the neurites of the BC models. Therefore, we increased the number of compartments to 139 and 152 for the OFF- and an ON-BC, respectively, which is sufficient to accurately represent all major neurites without becoming computationally too expensive (requiring approximately 20 min and 13 min per simulation for a 31 s light stimulus for the OFF- and ON-BC, respectively).

## Ion channels and synapses - literature review

The complement and distribution of voltage- or ligand-gated ion channels shapes the response of neurons. Here, ion channels are modeled as additional electrical elements in the compartments' membrane with conductances dependent on time varying parameters, such as the membrane potential and the calcium concentration within the cell. In addition to the equations that govern a channel's kinetics, their location in the cell has to be defined. After a literature review of retinal cone bipolar cell types in mice, we decided to model the OFF- and ON-type for which we could gather most information, namely BC3a and BC5. Currently, there are three accepted subtypes of BC5: 5o, 5i and 5t (*Greene et al., 2016*). Here, we modeled the BC5 subtype that expresses voltage-gated sodium channels (*Hellmer et al., 2016*) which probably also corresponds to the more transient BC5 subtype reported in *Ichinose et al., 2014*. The TTX sensitivity observed in *Matsumoto et al., 2019* suggests that both, 5o and 5i express voltage-gated sodium channels. To make our model consistent, we used data from the same BC5 subtype (5o) for the morphology, the target data and the number of cone contacts. A summary of all used channels, their location within the models and the respective references can be found in *Table 1*. The following paragraphs describe which channels were included in the models and why. Note, however, that for all channels (except the L-type calcium channel in the axon terminals, as calcium channels are necessary in the model for neurotransmitter release) channel densities of zero were included in the prior distributions, thereby allowing the parameter inference to effectively remove ion channels from the model.

In their axon terminals, cones express L-type calcium ($Ca_L$) channels that mediate release of the transmitter glutamate (*Morgans et al., 2005*; *Mansergh et al., 2005*; *Ingram et al., 2020*). We modeled calcium extrusion purely with calcium pumps ($Ca_P$) since other mechanisms such as sodium-calcium-exchangers probably only play a minor functional role in cones (*Morgans et al., 1998*). Additionally, there is evidence that cones express hyperpolarization-activated cyclic nucleotide-gated cation (HCN) channels of the type 1, mostly in the inner segment but also in the axon (*Knop et al., 2008*; *Van Hook et al., 2019*). The presence of $HCN_3$ channels in mouse cones is more

**Table 1.** Ion channels of biophysical models.

| Channel | Cone | OFF-BC (type 3a) | ON-BC (type 5) | Cone references | BC references |
|---|---|---|---|---|---|
| $Ca_L$ | AT | S, AT | S, AT | *Morgans et al., 2005*; *Mansergh et al., 2005* | *Van Hook et al., 2019* |
| $Ca_T$ | | S, AT | | | *Van Hook et al., 2019* |
| $Ca_P$ | AT | S, AT | S, AT | *Morgans et al., 1998* | *Morgans et al., 1998* |
| $HCN_1$ | All | | D, S, AT | *Knop et al., 2008*; *Van Hook et al., 2019* | *Knop et al., 2008*; *Hellmer et al., 2016* |
| $HCN_4$ | | D, S, AT | | | *Hellmer et al., 2016*; *Knop et al., 2008* |
| $K_V$ | IS/S | DD, PA, DA | DD, PA, DA | *Knop et al., 2008*; *Van Hook et al., 2019* | *Ma et al., 2005* |
| $K_{ir}$ | | S | S | | *Cui and Pan, 2008*; *Knop et al., 2008* |
| $Cl_{Ca}$ | AT | | | *Yang et al., 2008*; *Caputo et al., 2015* | |
| $Na_V$ | | DA | DA | | *Hellmer et al., 2016* |

Regions of ion channels and the respective abbreviations as in *Figure 1*.

D refers to the combination of DD and PD. All refers to the combination of IS/S, A, and AT.

If multiple regions are stated for a neuron, the ion channel density differs between them.

controversial. These channels have been observed in rat cones (*Müller et al., 2003*), and a more recent study also found evidence for $HCN_3$ channels at the synaptic terminals of mouse cones, but could not observe any functional differences between wild-type and $HCN_3$-knockout mice. To restrict the number of model parameters, we did not include $HCN_3$ in our cone model. However, we added calcium-activated chloride ($Cl_{Ca}$) channels to the axon terminals (*Yang et al., 2008*; *Caputo et al., 2015*) and voltage-gated potassium channels $K_V$ at the inner segment (*Van Hook et al., 2019*).

Our BC5 type expresses voltage-gated sodium ($Na_V$) channels at the axon shaft (*Hellmer et al., 2016*). Another study found inward-rectifier potassium ($K_{ir}$) channels at the soma of BC5 (*Knop et al., 2008*), which were also found in the homologous type in rat (*Cui and Pan, 2008*). Additionally, BC5 express HCN channels at the axon terminal, the soma and the dendrites (*Knop et al., 2008*; *Hellmer et al., 2016*). From the four subtypes of HCN, BC5 seem to almost exclusively express $HCN_1$. In the rat, there is also evidence for the expression of $HCN_4$ channels in BC5 (*Müller et al., 2003*; *Ivanova and Müller, 2006*), but this could not be verified for mice. Data from rat suggests that BCs with $Na_V$ channels also express $K_V$ channels (*Ma et al., 2005*). We therefore added $K_V$ channels at the dendrites and the axon.

Similar to BC5, BC3a express HCN channels at the axon terminals, the soma and the dendrites. However, instead of $HCN_4$ they express $HCN_4$ (*Hellmer et al., 2016*; *Knop et al., 2008*). There is also evidence that BC3a express $Na_V$ channels at the axon shaft (*Hellmer et al., 2016*), which were also found in the homologous type in rat (*Cui and Pan, 2008*). Just like for BC5, we added also $K_V$ in BC3a were only reported for rat so far (*Cui and Pan, 2008*). As we could not find any evidence for the lack of $K_{ir}$ channels in mouse BC3a and the channel repertoires of BC3a in mouse and rat are overall very consistent, we included them in our model.

The distribution of calcium channels in mouse CBCs is largely unknown (*Van Hook et al., 2019*). In the rat retina, there is evidence for T-type calcium ($Ca_T$) channels in BC3a (*Ivanova and Müller, 2006*). Calcium currents of unspecified type were observed in BC5 (*Cui and Pan, 2008*). Generally, L-type calcium ($Ca_L$) channels are believed to mediate neurotransmitter release in almost all BCs across types and species (*Van Hook et al., 2019*). Therefore, we included them in both BC models. The literature review in *Van Hook et al., 2019* suggests that T-type calcium channels might be exclusively expressed in BC3. In mouse BC3b, the simultaneous expression of both $Ca_T$ and $Ca_L$ has been described (*Cui et al., 2012*). Furthermore, the latter and other studies (*Hu et al., 2009*; *Satoh et al., 1998*) suggest that voltage-gated calcium channels might not be located in the axon terminals only, but also in the soma and might play a role in signal transmission within the cell. Based on the studies mentioned, we assumed that BC3a and BC5 express $Ca_L$ in the axon terminals and potentially also at the soma. The BC3a model may additionally use $Ca_T$ channels, both at the soma and at the axon terminals. For calcium extrusion, we added calcium pumps (*Morgans et al., 1998*).

BC5 receive input from cones via the metabotropic glutamate receptor 6 (mGluR6) (*Van Hook et al., 2019*). BC3a receive input from cones via kainate receptors (*Ichinose and Hellmer, 2016*). We modeled the kainate receptors by modifying the inactivation time constant $\tau_\gamma$ of the AMPA receptors included with NeuronC.

## Ion channels and synapses - implementation

All ion channels in this study were based on the models available in NeuronC. We used both Hodg-kin-Huxley (HH) and Markov-Sequential-State (MS) channel implementations. Since we did not add channel noise to our model, every HH channel could have also been described as an equivalent MS channel. However, since HH channels are computationally less expensive, we used HH implementations wherever possible. Implementation details and references are listed in *Table 2*. The L-type calcium channel, for example, was based on the HH model defined by the following equations:

$$\frac{1}{r_e} := g_e = c^3 \cdot g_{max}, \qquad \frac{\delta}{\delta t} c = (1-c) \cdot \alpha(V) - c \cdot \beta(V), \qquad (4)$$

$$\alpha(V) = \eta_T \cdot \frac{-0.04 \cdot (V+15)}{\exp(-0.04 \cdot (V+15)) - 1} \cdot \frac{1}{\text{ms}}, \qquad \beta(V) = \eta_T \cdot 5 \cdot \exp\left(\frac{V+38}{-18}\right) \cdot \frac{1}{\text{ms}}. \qquad (5)$$

Here, $\eta_T$ corrects for differences between the temperature of the simulated cell $T_{sim}$ and the

**Table 2.** Ion channel implementation details and optimized channel parameters.

| Channel | NeuronC | Type | States | Parameters | Channel remarks and references |
|---|---|---|---|---|---|
| Kainate rec. | AMPA1 | MS | 7 | $STC$, $\tau_\gamma$ | Based on *Jonas et al., 1993*. |
| mGluR6 | mGluR | | | $STC$ | See NeuronC documenation. |
| $Ca_L$ | CA0 | HH | (4) | $\Delta V_\alpha$, $\tau_\alpha$ | Based on *Karschin and Lipton, 1989*. |
| $Ca_T$ | CA7 | MS | 12 | $\Delta V_\alpha$, $\tau_\alpha$ | Modification of *Lee et al., 2003*. |
| $Ca_P$ | | | | $Ca_{PK}$ | See NeuronC documenation. |
| $HCN_{1/2/4}$ | K4 | MS | 10 | | Based on *Altomare et al., 2001*. |
| $K_V$ | K0 | HH | (5) | $\Delta V_\alpha$, $\tau_\alpha$ | Based on *Hodgkin and Huxley, 1952*. |
| $K_{ir}$ | K5 | MS | 3 | $\Delta V_\alpha$ | Modification of *Dong and Werblin, 1995*. |
| $Cl_{Ca}$ | CLCA1 | MS | 12 | | Modification of *Hirschberg et al., 1998*. |
| $Na_V$ | NA5 | MS | 9 | $\Delta V_\alpha$, $\Delta V_\gamma$, $\tau_{all}$ | Based on *Clancy and Kass, 2004*. |

temperature for which the channel equations were defined $T_{eq}$ based on a temperature sensitivity $Q_{10}$ which can vary between ion channels and state transitions:

$$\eta_T = \exp\left(\log\left(Q_{10}\right) \cdot \left(T_{sim} - T_{eq}\right) / 10\,\mathrm{K}\right). \tag{6}$$

There are several sources for model uncertainty about the exact channel kinetics. First, not all channel models used here were developed based on mouse data resulting in species-dependent differences. Second, we do not always know the exact subtypes of ion channels, for example in the case of the T-type calcium channel. Third, the exact temperature sensitivities $Q_{10}$ are not known. Therefore, we estimated transition rates and thresholds for state transitions during the parameter inference. For this, we allowed for offsets $\Delta V$ relative to $V$ in the rate equations and additionally, we estimated relative time constants $\tau$ for the rates. For example *Equation 4* was changed to:

$$\frac{\delta}{\delta t}c = (1-c) \cdot \frac{1}{\tau_\alpha} \cdot \alpha(V-_\alpha) - c \cdot \frac{1}{\tau_\beta} \cdot \beta(V - \Delta V_\beta). \tag{7}$$

To keep the parameter space as small as possible, we only optimized the kinetics of ion channels with high uncertainty (e.g. $K_V$) or with high relevance for the exact timing of the neurotransmitter release (e.g. $Ca_L$ and $Ca_T$). Additionally, we constrained the channel parameters to physiologically plausible ranges. *Table 2* summarizes which channel parameters were estimated during parameter optimization. Time constants $\tau$ and voltage offsets $\Delta V$ not optimized were set to one and zero, respectively. For the $Na_V$, a single time constant $\tau_{all}$ was used to modify all time constants proportionally. The calcium pump dynamics were modified by changing the calcium concentration $Ca_{PK}$ that causes half of the maximum calcium extrusion velocity. The BC glutamate receptors were optimized by allowing for a change in the synaptic transmitter concentration at the receptors by a factor of $STC$, which might be smaller for the OFF-BC than for the ON-BC given the greater distance between the release sites of the cones and the dendritic tips of the BCs (*Behrens et al., 2016*). The simulated cell temperature $T_{sim}$ was set to 37°C if not stated otherwise. For further information we refer to the NeuronC documentation (*Smith, 1992*).

## Neurotransmitter release

The glutamate release of cones and BCs release is mediated through ribbon synapses that release vesicles in response to calcium influx in a nonlinear way (*Matthews and Fuchs, 2010*; *tom Dieck and Brandstätter, 2006*; *Baden et al., 2013*). We modeled the ribbon synapses with a standard model (*Smith, 1992*) including a readily releasable pool (RRP) from which vesicles can be released (*Lagnado and Schmitz, 2015*). The presence of multiple release pools shapes the dynamic of release at the ribbon synapse and make it state dependent, allowing for rapid adaptational processes at the synaptic site (*Baden et al., 2013*). In the model, the current release rate is dependent on the number of vesicles currently available $v_{RRP}$ in the RRP, the maximum number of vesicles $v_{RRP}^{max}$ in the RRP and the intracellular calcium concentration $[Ca]$. In NeuronC, calcium is modeled in radial shells through

which calcium can diffuse deeper into the neuron. For the release of neurotransmitter, only the calcium concentration in the first shell $[Ca]_0$ (equivalent to the concentration at the membrane) is considered. The release rate $r$ is computed as:

$$r(t) = \left( [Ca]_0(t) \frac{1e6}{\text{mol}} \right)^2 \cdot \frac{v_{RRP}(t)}{v_{RRP}^{max}} \cdot g_l \cdot \frac{vesicles}{s}, \tag{8}$$

where $g_l$ is a linear gain factor. $g_l$ and $v_{RRP}^{max}$ were optimized for every cell type individually. The RRP is constantly replenished with a constant rate that is equivalent to the maximum sustainable release rate $r_{msr}$. At a time $t$, for a simulation time step $\Delta t$, the vesicles in the pool are updated as follows:

$$v_{RRP}(t + \Delta t) = v_{RRP}(t) - r(t) \cdot \Delta t + r_{msr} \cdot \Delta t \cdot \left( 1 - \frac{v_{RRP}(t)}{v_{RRP}^{max}} \right). \tag{9}$$

For the cone model, $r_{msr}$ was set to 100 vesicles per second based on *Berntson and Taylor, 2003*. The prior for $v_{RRP}^{max}$ was based on RRP sizes reported for salamander (*Thoreson et al., 2016*; *Bartoletti et al., 2010*). For the BCs, $r_{msr}$ was set to eight vesicles per second based on the reported value for rat rod bipolar cells in *Singer and Diamond, 2006*. The prior for $v_{RRP}^{max}$ was based on *Wan and Heidelberger, 2011*.

## Bayesian inference for model parameters

To estimate the free parameters of the multicompartment models, we used a Bayesian likelihood-free inference framework called Sequential Neural Posterior Estimation (SNPE) (*Lueckmann et al., 2017*; *Gonçalves et al., 2020*). The goal of the parameter estimation was to find parameter regions for which the model outputs match the experimentally observed glutamate release in response to a light stimulus. Details of the target data, the stimulus, the comparison between experimental and simulated data and the inference algorithm are described below. To be able to simulate the light response of the BC models, we inferred the parameters of the cone model first.

### Target data of neuron models

As target data, we used two-photon imaging data recorded with an intensity-based glutamate-sensing fluorescent reporter (iGluSnFR) (*Marvin et al., 2013*). All animal procedures were approved by the governmental review board (Regierungspräsidium Tübingen, Baden-Württemberg, Konrad-Adenauer-Str. 20, 72072 Tübingen, Germany) and performed according to the laws governing animal experimentation issued by the German Government.

To constrain the cone models, we used glutamate traces of two cone axon terminals (Figure 5—figure supplement 1A) in response to a full-field chirp light stimulus (Figure 5A). The traces were recorded in one transgenic mouse (B6;129S6-Chat$^{tm2(cre)Lowl}$J, JAX 006410, crossbred with *Gt(ROSA) 26Sor$^{tm9(CAG-tdTomato)Hze}$*, JAX 007905) that expressed the glutamate biosensor iGluSnFR ubiquitously across all retinal layers after intravitreal injection of the viral vector AAV2.7m8.hSyn.iGluSnFR (provided by D. Dalkara, Institut de la Vision, Paris). The cone glutamate release in the outer plexiform layer was recorded in x-z scans (64 × 56 pixels at 11.16 Hz; *Zhao et al., 2019*). Region-of-interest (ROIs) were drawn manually and traces of single ROIs were then normalized and upsampled to 500 Hz as described previously (*Franke et al., 2017*; *Szatko et al., 2019*). For each axon terminal, we computed the mean over five traces. Both means were then aligned by minimizing the mean squared error between them, and the mean of the two aligned means was used as target data for the cone model (Figure 5—figure supplement 1B).

For the BC models, we used mean glutamate traces of BC3a (n = 19 ROIs) and BC5o (n = 13 ROIs) (Figure 5—figure supplement 1C–F) in response to a chirp light stimulus (Figure 6A) from a recently published dataset (*Franke et al., 2017*). In that study, glutamate responses were recorded from BC terminals at different depths of the inner plexiform layer (x-y scans, 64 × 16 pixels at 31.25 Hz). ROIs were drawn automatically based on local image correlation and traces of single ROIs were normalized and upsampled to 500 Hz (see above). Since we simulated isolated BCs (except for the cone input), we used the responses to a local 'chirp' light stimulus recorded with the glycine receptor blocker strychnine, which means that the target data is less affected by inhibition from small-field amacrine cells. We did not consider input from GABAergic, wide-field amacrine cells, because these are not strongly activated by the local chirp stimulus (*Franke et al., 2017*). The shape of the BC

stimulus differed from the cone stimulus as contrast was not linearized for the BC recordings and therefore intensity modulations below 20% brightness were weakly rectified.

## Light stimulus and cell response

We first matched the experimental with the simulated stimulus. For this, we used the digital stimuli and corrected both timing and amplitude (using a sigmoid function) to minimize the mean squared error with respect to the experimentally recorded stimuli, correcting for delays and non-linearities in the displaying process. Then we linearly transformed the light stimulus such that the simulated photon absorption rates were $10 \times 10^3 \, \mathrm{P}^*/(\mathrm{s} \cdot \mathrm{cone})$ for the lowest and $31 \times 10^3 \, \mathrm{P}^*/(\mathrm{s} \cdot \mathrm{cone})$ for the highest stimulus intensity including the background illumination, approximating the values reported in *Franke et al., 2017*. In NeuronC, the photon absorption rate acts as input to a phototransduction model (*Nikonov et al., 1998*), which provides the hyperpolarizing current entering the inner segment. The membrane potential in the axon terminal compartment regulates the calcium influx into the cell which in turn influences the glutamate release rate. This glutamate release from the simulated cones modifies the opening probability (the fraction of open channels in the deterministic case) of postsynaptic receptors, which drive the BC models.

## Discrepancy function

To compare model outputs to the experimentally observed target data, we defined a discrepancy function $\delta$. Since the target traces were relative fluorescence intensities, the absolute number of released glutamate vesicles could not be directly inferred from the target data, and the data only constrained relative variations in the release rate during simulation. Because we also wanted to constrain our models to plausible membrane potentials and release rates, we combined the following seven discrepancy measures:

- $\delta_{iGluSnFR}$: The mean squared error between the experimental and simulated iGluSnFR trace.
- $\delta_{Rate}^{Rest}$: A penalty for implausibly high resting release rates.
- $\delta_V^{Rest}$: A penalty for implausibly low or implausible high resting membrane potentials.
- $\delta_{Rate}^{\Delta}$: A penalty for implausibly low release rate changes.
- $\delta_V^{\Delta}$: A penalty for implausibly low membrane potential changes.
- $\delta_V^{min}$: A penalty for implausibly low membrane potentials.
- $\delta_V^{max}$: A penalty for implausibly high membrane potentials.

The discrepancy between a model output $m$ and the target data $\nu_t$ was computed as:

$$\delta(\nu_t, m) = [\delta_{iGluSnFR}(\nu_t, m), \delta_{Rate}^{Rest}(m), \delta_V^{Rest}(m), \delta_{Rate}^{\Delta}(m), \delta_V^{\Delta}(m), \delta_V^{min}(m), \delta_V^{max}(m)]^{\top}. \tag{10}$$

To identify the overall 'best' samples, we computed the total discrepancy as the absolute-value norm of the discrepancy vector: $\delta_{tot}(\nu_t, m) = \|\delta(\nu_t, m)\|$.

The discrepancy function $\delta_{iGluSnFR}$ (*Equation 10*) computes the distance between a simulated iGluSnFR trace $\nu_m$ and an iGluSnFR target $\nu_t$. To estimate the simulated iGluSnFR trace, we convolved the glutamate release rate $\mathbf{r}_m$ with an iGluSnFR kernel $\kappa$. Here, the time-dependent kernel function $\kappa$ was approximated with an exponential decay function, based on iGluSnFR intensity changes to spontaneous vesicle release reported in *Marvin et al., 2013*:

$$\nu_m = \mathbf{r}_m * \kappa, \qquad \kappa(t) = \exp(-t/60\,\mathrm{ms}). \tag{11}$$

The discrepancy was then computed as the euclidean distance between the simulated and the target iGluSnFR trace with respect to a distance minimizing linear transformation of the simulated trace. This linear transformation was necessary because the target traces only reflect relative fluorescence changes. The discrepancy was normalized to be between zero and one by dividing by the variance $\|\nu_t - \mu_t\|^2$, where $\mu_t$ is the mean of the target data.

$$\delta_{iGluSnFR}(\nu_t, \nu_m) = \min_{a,b} \frac{\|\nu_t - (a + b \cdot \nu_m)\|^2}{\|\nu_t - \mu_t\|^2}, \qquad b \geq 0. \tag{12}$$

For all other discrepancies, specific values of the glutamate release rate (in the case of the BCs, the mean release rate over all synapses) or the somatic membrane potential were compared to a

lower and an upper bound of target values $t_l$ and $t_u$, such that values within these bounds were assigned a discrepancy of 0.0. Outside this range, the discrepancy was defined by additional bounds $p_l$ and $p_u$. Given a specific value of the simulation $y_m$, the respective discrepancy $\delta_\bullet^\bullet(y_m)$ was computed as:

$$\delta_\bullet^\bullet(y_m) = \begin{cases} -1 + \exp\left(-2\frac{(y_m - t_l)^2}{(t_l - p_l)^2}\right), & \text{if } y_m \in (-\infty, t_l), \\ 0, & \text{if } y_m \in [t_l, t_u], \\ 1 - \exp\left(-2\frac{(y_m - t_u)^2}{(t_u - p_u)^2}\right), & \text{if } y_m \in (t_u, \infty). \end{cases} \tag{13}$$

To compute $\delta_{Rate}^{Rest}$ and $\delta_V^{Rest}$, the resting release rate $r_m^0$ and resting membrane potential $v_m^0$ for the background light adapted state were extracted. For the BC models, the resting membrane potential was not penalized for values between $t_l = -65\,\text{mV}$ and $t_u = -45\,\text{mV}$ based on reported values for mice (**Ichinose et al., 2014**; **Ichinose and Hellmer, 2016**) and rat retina (**Ma et al., 2005**). For the cone model, the expected resting membrane potential was more depolarized between $t_l = -55\,\text{mV}$ and $t_u = -40\,\text{mV}$(**Cangiano et al., 2012**).

The discrepancy of the resting release rate $\delta_{Rate}^{Rest}$ was computed similarly. For the BC models, the lower bound $t_l$ was set to zero. As mentioned earlier, we limited our BC models to have a maximum sustainable release rate of 8 vesicles per second based on **Singer and Diamond, 2006**. We allowed non-zero resting release rates due to the background light and spontaneous vesicle fusion but constrained it to values lower than the maximum sustainable release rate (**Kavalali, 2015**; **Baden et al., 2014**). For the OFF-BC we chose an upper bound of 4 vesicles per second (half the maximum sustainable release rate). For the ON-BC, we chose a slightly smaller value of 3 vesicles per second. This difference was based on the observation that the ON-BC target never falls significantly below the value of the resting state, indicating that the resting release rate is probably close to zero and can therefore not become smaller. In contrast, the OFF-BC target falls below the resting value right after stimulus onset, indicating a small but non-zero resting release rate. For the cone model, we assumed a comparably high resting release rate between $t_l = 50$ and $t_u = 80$ vesicles per second based on the assumed higher maximum sustainable release rate and the fact that cones show steady release in darkness (**Choi et al., 2005**; **Sheng et al., 2007**).

For the penalty on implausible release changes $\delta_{Rate}^\Delta$, we computed the largest absolute difference $\Delta r$ between the resting release rate $r_m^0$ and release rates $\mathbf{r}_m$ after stimulus onset. $\delta_V^\Delta$ was computed analogously but for the membrane potential $\mathbf{v}_m$ and the resting membrane potential $v_m^0$:

$$\Delta r = \max|\mathbf{r}_m - r_m^0| \quad \text{and} \quad \Delta v = \max|\mathbf{v}_m - v_m^0|. \tag{14}$$

$\delta_{Rate}^\Delta(y_m)$ and $\delta_V^\Delta(y_m)$ were then computed by using the differences $y_m = \Delta r$ and $y_m = \Delta v$, respectively, in **Equation 13**. For the BC release rate, we did not penalize differences larger than $t_l = 5$ vesicles per second. For the cone, we expected much larger differences between $t_l = 50$ to $t_u = 65$ vesicles per second due to their larger maximum sustainable release rate. For the membrane potential, we expected a difference of at least $t_l = 5\,\text{mV}$ based on light step responses recorded with patch clamp in mouse BCs (**Ichinose et al., 2014**; **Ichinose and Hellmer, 2016**). Since here, the stimulus contrast was higher, we only used the reported values as lower bounds but allowed the model to have larger variation, namely up to $t_u = 25\,\text{mV}$ for the OFF- and $t_u = 15\,\text{mV}$ for the ON-BC. We allowed greater membrane potential variation in the OFF-BC, because it receives input from more cones.

For the discrepancy measures $\delta_V^{min}$ and $\delta_V^{max}$, we computed the minimum and maximum of the membrane potential $\mathbf{v_m}$ after stimulus onset and used again **Equation 13**. For $\delta_V^{min}$, we chose $t_l = -80\,\text{mV}$ for the BCs and $t_l = -60\,\text{mV}$ for the cone model, and in both cases $t_u = \infty$. For $\delta_V^{max}$, we chose $t_u = -10\,\text{mV}$ for the BCs and $t_u = -35\,\text{mV}$ for the cone, and in both cases we set $t_l = -\infty$. The BC values are based on data from rat (**Ma et al., 2005**) and ground squirrel (**Saszik and DeVries, 2012**); the cone values are based on **Cangiano et al., 2012**.

All values for $p_l$ and $p_u$ were based on pilot simulations with the goal to distribute the penalties where they most mattered. All discrepancies (except for $\delta_{iGluSnFR}$) and their respective values $p_l$, $p_u$, $t_l$ and $t_u$ are illustrated in **Figure 2** for clarity and summarized in **Table 3**.

## Priors

The inference method SNPE is a Bayesian method and therefore it needs a prior distribution $p(\theta)$ for the parameters $\theta$ to estimate the posterior. We chose truncated normal distributions for all priors because they allow for weighting of more plausible parameters (in contrast to e.g. uniform distributions), while they enable restrictions to plausible ranges (in contrast to e.g. normal distributions). A $d$-dimensional truncated normal distribution $\mathcal{N}_T$ is defined by a mean $\boldsymbol{\mu} = (\mu_1, ..., \mu_d)^T$, a $d \times d$ covariance matrix $\Sigma$ and a $d$-dimensional space $W = [a_1, b_1] \times ... \times [a_d, b_d]$:

$$\mathcal{N}_T(\theta|\boldsymbol{\mu}, \Sigma, W) = \begin{cases} \frac{\exp\left(-0.5(\theta-\boldsymbol{\mu})^T\Sigma^{-1}(\theta-\boldsymbol{\mu})\right)}{\int_W \exp\left(-0.5(\omega-\boldsymbol{\mu})^T\Sigma^{-1}(\omega-\boldsymbol{\mu})\right)d\omega} & \text{if} \quad \theta \in W, \\ 0 & \text{otherwise.} \end{cases} \quad (15)$$

The prior means $\mu_i$ and truncation bounds $[a_i, b_i]$ were based on experimental data wherever possible (see *Appendix 1—table 1* and *2*), including data from rat and different cell types such as rod bipolar cells, as well as pilot simulations. For parameter inference, we normalized the parameter space such that the truncation bounds were [0, 1] in all dimensions. The diagonal entries of the prior covariance matrix $\Sigma$ were set to $0.3^2$. Because it is difficult to find prior knowledge about the dependencies of parameters, we set all non-diagonal entries to zero. To sample from $\mathcal{N}_T$, we implemented a rejection sampler, that samples from a normal distribution with the same mean $\boldsymbol{\mu}$ and covariance matrix $\Sigma$ and resamples all $\theta$ not in $W$.

### Inference algorithm

SNPE estimates a posterior parameter distribution represented by a mixture-density network, based on sampling, that is, model evaluations for randomly drawn parameters. Inference is performed in several rounds. In every round $j$, the algorithm draws $N$ parameters from a sampling distribution $\tilde{p}_j(\theta)$ to estimate the posterior distribution $p(\theta|\mathbf{x}_{target})$, where $\mathbf{x}_{target}$ is a summary statistic of the target data.

In the first round, parameter samples $\theta_n$ are drawn from the prior, that is, $\tilde{p}_1(\theta) = p(\theta)$, and the multicompartment model is evaluated for all $\theta_n$. From each simulated response, a summary statistic $\mathbf{x}_n$ is computed, resulting in $N$ pairs of parameters and summary statistics $(\theta_n, \mathbf{x}_n)$. At the end of the round, a mixture-density network is trained with summary statistics $\mathbf{x}$ as input, and the parameters $\phi$

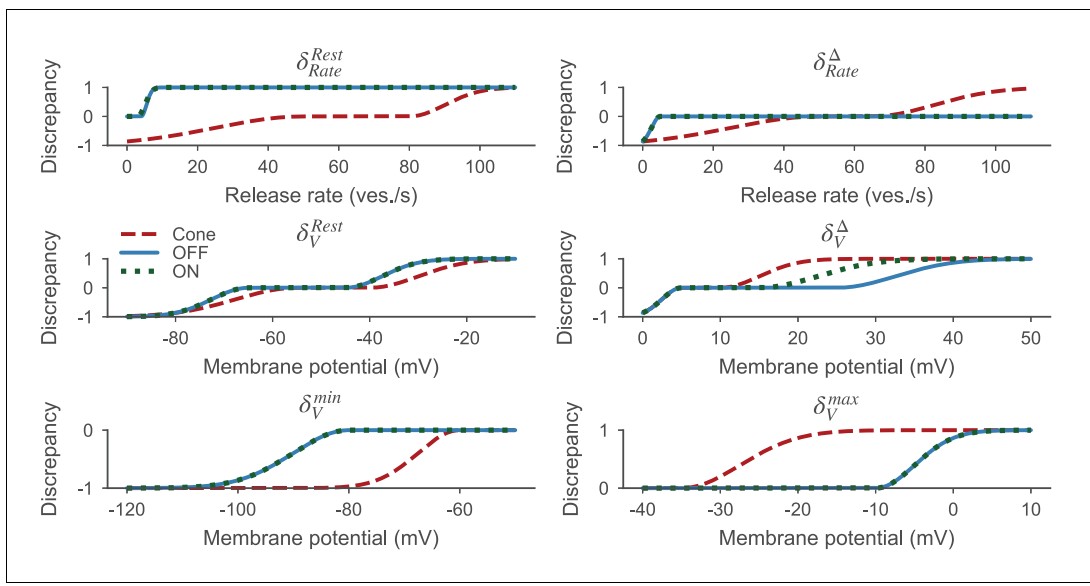

**Figure 2.** Discrepancy measures based on *Equation 13* for the cone (red dashed line), the OFF- (blue solid line) and ON-BC (green dotted line). The parameters defining the discrepancy measures are listed in *Table 3*. All discrepancy measures are between zero and one per definition.

**Table 3.** Parameters of discrepancy measures.

| $\delta$: | Cone | | | | BC (3a \| 5o) | | | | References |
|---|---|---|---|---|---|---|---|---|---|
| | $p_l$ | $t_l$ | $t_u$ | $p_u$ | $p_l$ | $t_l$ | $t_u$ | $p_u$ | |
| $\delta_{Rate}^{Rest}$ | 0 | 50 | 80 | 100 | 0 | 0 | 3 \| 4 | 7 | *Choi et al., 2005*; *Sheng et al., 2007*; *Berntson and Taylor, 2003*; *Singer and Diamond, 2006* |
| $\delta_{V}^{Rest}$ | −80 | −55 | −40 | −20 | −80 | −65 | −45 | −30 | *Cangiano et al., 2012*; *Ichinose et al., 2014*; *Ichinose and Hellmer, 2016*; *Ma et al., 2005* |
| $\delta_{Rate}^{\Delta}$ | 0 | 50 | 65 | 100 | 0 | 5 | $\infty$ | $\infty$ | |
| $\delta_{V}^{\Delta}$ | 0 | 5 | 10 | 20 | 0 | 5 | 15 \| 25 | 40 | *Ichinose et al., 2014*; *Ichinose and Hellmer, 2016* |
| $\delta_{V}^{min}$ | −75 | 60 | $\infty$ | $\infty$ | −100 | −80 | $\infty$ | $\infty$ | *Cangiano et al., 2012*; *Ma et al., 2005*; *Saszik and DeVries, 2012* |
| $\delta_{V}^{max}$ | −$\infty$ | −$\infty$ | −35 | −20 | −$\infty$ | −$\infty$ | −10 | 0 | *Cangiano et al., 2012*; *Ma et al., 2005*; *Saszik and DeVries, 2012* |

of a mixture of Gaussian distribution $q_\phi(\theta|\mathbf{x})$ as output. The network is trained by minimizing the loss function $\mathcal{L}$:

$$\mathcal{L}(\phi) = -\frac{1}{N} \sum_n^N \frac{p(\theta_n)}{\tilde{p}(\theta_n)} K(\mathbf{x}_n) \log(q_\phi(\theta_n|\mathbf{x}_n)), \tag{16}$$

where $K$ is a kernel function with values between zero and one that weights the influence of samples on the network training. $K(\mathbf{x}_n)$ is close to one for samples with summary statistics $\mathbf{x}_n$ close to the the target summary statistic $\mathbf{x}_{target}$ and becomes smaller with increasing discrepancy between $\mathbf{x}_n$ and $\mathbf{x}_{target}$. This means, the network tries to find parameter distributions $q_\phi(\theta|\mathbf{x})$ that describe the distribution of samples for any given summary statistic $\mathbf{x}$. Or, in other words, the network is trained to find a mapping from summary statistics to parameter distributions. $K$ ensures that the network focuses its capacity on summary statistics close to the target summary statistic. After training the network, it can be evaluated at a summary statistic $\mathbf{x}^*$ to obtain the posterior parameter distributions for the given summary statistic. Evaluating at $\mathbf{x}^* = \mathbf{x}_{target}$ yields an approximation of the true posterior distribution $p(\theta|\mathbf{x}_{target}) \approx q_\phi(\theta|\mathbf{x}_{target})$. This posterior can either be used as the sampling distribution of the next round $\tilde{p}_{j+1}(\theta)$, or—if the algorithm is stopped—as the final posterior distribution. The relative probability $p(\theta_n)/\tilde{p}(\theta_n)$ in *Equation 16* weights samples not drawn from the prior, which ensures that Bayes's rule is not violated. A detailed proof that this actually yields an approximation of the true posterior in the Bayesian sense can be found in *Lueckmann et al., 2017*.

We based our algorithm on the Python code available at https://github.com/mackelab/delfi version 0.5.1 (*Macke Lab, 2020*) with the following settings and modifications. We modeled $q_\phi$ as a single Gaussian, because we noticed that mixture of Gaussians almost always collapsed to a single component after a few rounds. Both, intermediate and final posteriors were truncated using the truncation bounds of the prior. The truncation was performed after network training. For every neuron model, we drew $N = 2,000$ samples per round and stopped the algorithm after the fourth round. Two hundred additional samples were drawn from the posterior for further analysis. Since we wanted to use the posterior samples to simulate the effects of electrical stimulation on the BCs, the number of compartments was increased in this last step to 139 and 152 for the OFF- and ON-BC, respectively.

As summary statistics of model outputs $m_n$, we used the discrepancy function $\delta(\nu_t, m_n) = \mathbf{x}_n$ (see *Equation 10*), which describes the discrepancy between model outputs and the target data. The target summary statistic was set to be a zero-vector $\mathbf{x}_{target} = [0, ..., 0]^\top$, since the target should have a discrepancy of zero with respect to itself. The first dimension of $\delta$, $\delta_{iGluSnFR}$, computes the distance between the simulated and experimentally observed iGluSnFR trace. Considering the noise in the target data, observing a discrepancy of zero is virtually impossible. Therefore, evaluating the network at $\mathbf{x}_{target} = [0, ..., 0]^\top$ is based on extrapolation, that is, the mixture-density network is evaluated for an input where it was not trained on. This, as we observed during pilot experiments, often led to posterior estimates of poor quality or endless loops of resampling. So instead of evaluating the

network at $\mathbf{x}_{target} = [0, ..., 0]^\top$ to obtain the posterior estimate, the network was instead evaluated at $\mathbf{x}_{target} = [x_{min}^{iGluSnFR}, 0, ..., 0]^\top$, where $x_{min}^{iGluSnFR}$ is the the smallest $\delta_{iGluSnFR}$ sampled during this round. This is roughly equivalent to assuming that the best strategy for extrapolation is to simply use the estimate at the boundary. For the weighting function $K$, we used zero-centered Gaussian kernels $k$ with a bandwidth of $\sigma = 0.25$ in all dimensions but the first one. In the first dimension, that is the weighting kernel for $\delta_{iGluSnFR}$, we also used an adaptive strategy and both, the mean $\mu_{iGluSnFR}$ and the bandwidth $\sigma_{iGluSnFR}$ of the kernel, were updated in every round:

$$\mu_{iGluSnFR} = x_{min}^{iGluSnFR}, \qquad \sigma_{iGluSnFR} = q_{20}^{iGluSnFR} - x_{min}^{iGluSnFR}, \qquad (17)$$

where $q_{20}^{iGluSnFR}$ is the 20th percentile of all sampled iGluSnFR discrepancies of the same round. $K$ was computed as the product of all scalar kernels $k$.

Some parameter combinations caused the neuron simulation to become numerically unstable. If a simulation could not successfully terminate for this reason, the sample was ignored during training of the mixture-density network by setting the kernel weight to zero. In other cases, the BC models had a second, strongly depolarized and therefore biologically implausible equilibrium state. To test for this, we simulated a somatic voltage clamp to 30 mV for 100 ms and checked whether the membrane potential would recover to a value of $-30$ mV or lower within additional 300 ms. Samples not recovering to $\leq$ -30 mV were also ignored during training.

## Data analysis of simulated traces

The distance function $\delta_{iGluSnFR}(\nu_t, \nu_m)$ (see *Equation 12*) was used not only to compute the discrepancy between simulations and the respective targets but also more generally to compare different experimental and simulated iGluSnFR traces. The distance between two iGluSnFR traces $\nu_1$ and $\nu_2$ was computed as $\delta_{iGluSnFR}(\nu_1, \nu_2)$.

To quantify the timing precision of our neuron models, we estimated peak times in simulated and target iGluSnFR traces to compute pairwise peak time differences. For every peak in the simulated trace, we computed the time difference to the closest peak of the same polarity (positive or negative) in the target. We did not consider peaks between 16 s and 23 s of the stimulation for the cone and between 16 s and 21 s for the BC models, because the targets were to noisy for precise peak detection in these time windows. This resulted in approximately 35 positive and negative peak time differences per trace.

## Simulation of electrical stimulation

To simulate external electrical stimulation of our BC models, we implemented a two-step procedure. In the first step, the electrical field is estimated as a function of space and time across the whole retina for a given stimulation current. By setting a position of the BC multicompartment models within the retina, the extracellular voltage for every compartment can be extracted. In the second step, the extracellular voltages are applied to the respective compartments (*Figure 1C*) to simulate the neural response in NeuronC. To be able to perform the first step, we estimated the electrical properties of retinal tissue first. For this, we utilized the same algorithm that was used for parameter inference of the neuron models. To validate the framework, we simulated the electrical stimulation in *Corna et al., 2018* and compared experimental and simulated neural responses. Finally, we utilized the framework to find electrical stimuli for selective stimulation of OFF- and ON-bipolar cells. Details of the implementation and the experimental data are described in the following.

### Computing the extracellular voltage

We estimated the electrical field in the retina for a given electrical stimulus with the finite-element method using the software COMSOL Multiphysics (*Comsol, 2019*). We modeled the photoreceptor degenerated retina as a cylinder with a radius of 2 mm and a height of 105 μm (*Pennesi et al., 2012*). The stimulation electrodes were modeled as flat disks on the bottom of the retina. Above the retina, an additional cylinder with the same radius and a height of 2 mm was placed to model the electrolyte. The top of this cylinder was assumed to be the return electrode. The implementation of such a model with the subdivision into finite elements is shown in *Figure 3*. For a single circular stimulation electrode, the model was radially symmetric and could therefore be reduced to a half cross-

**Figure 3.** Model for the external electrical stimulation of the retina. (A) Schematic figure of the experimental setup for subretinal stimulation of ex vivo retina combined with epiretinal recording of retinal ganglion cells. Schematic modified from *Corna et al., 2018*. (B,C) Model for simulating the electrical field potential in the retina in 3D and 2D, respectively. The retina (darker blue) and the electrolyte above (lighter blue) are modeled as cylinders. The shown 3D model is radially symmetric with respect to the central axis (red dashed line). Therefore, the 3D and 2D implementations are equivalent, except that the computational costs for the 2D model are much lower. The 2D implementation is annotated with parameters that were either taken from the literature or inferred from experimental data. (D) Electrical field potential in the retina for a constant stimulation current of 0.5 µA for a single stimulation electrode with a diameter of 30 µm. Additionally, the compartments (black circles with white filling) of the ON-BC model are shown. The stimulation is subretinal meaning that the dendrites are facing the electrode (horizontal black line on bottom).

section as shown in *Figure 3* to increase the simulation speed without altering the results. The following initial and boundary conditions were applied to the model. The initial voltage was set to zero at every point $V(x, y, z, t = 0) = 0$. The surface normal current density $j_{stim}^{\perp}$ of stimulation electrodes was always spatially homogeneous and dependent on the total stimulation current $i_{stim}$ and the total surface area of all electrodes $A_{electrode}$:

$$j_{stim}^{\perp} = \frac{i_{stim}}{A_{electrode}}. \tag{18}$$

The potential of the return electrode was kept constant $V_{\text{return}}(t) = 0$. At all other boundaries, the model was assumed to be a perfect insulator $j_{other}^{\perp} = 0$. We assumed a spatially and temporally homogeneous conductivity and permittivity in both the retina and the electrolyte. The conductivity of the electrolyte was set to $\sigma_{\text{ames}} = 1.54 \, \text{S/m}$ based on *Eickenscheidt and Zeck, 2014* and its relative permittivity was assumed to be $\varepsilon_{ames} = 78$, based on the value for water. The conductivity $\sigma_{retina}$ and relative permittivity $\varepsilon_{retina}$ of the retina were optimized with respect to experimental target data as described below.

## Target data to infer the electrical parameters of the retina

To estimate the electrical properties of the retina, we first recorded target data. All procedures were approved by the governmental review board (Regierungspräsidium Tübingen, Baden-Württemberg, Konrad-Adenauer-Str. 20, 72072 Tübingen, Germany, AZ 35/9185.82–7) and performed according to the laws governing animal experimentation issued by the German Government. We applied different sinusoidal stimulation voltages $v_{stim}$ and recorded the evoked currents. Currents were recorded with ($i_{retina}^{rec}$) and without ($i_{ames}^{rec}$) retinal tissue placed on the micro-electrode array. In both cases, the recording chamber was filled with an electrolyte (Ames' medium, A 1420, Sigma, Germany). A single Ag/AgCl pellet (E201ML, Science Products) was used as a reference electrode and located approximately 1 cm above a customized micro-electrode array. The electrodes, made of sputtered iridium oxide had diameters of 30 µm and center-to-center distance of 70 µm. The stimulation current was calculated from the voltage drop across a serial 10 resistor in series with the Ag/AgCl electrode (*Corna et al., 2018*). The voltage drop was amplified using a commercial voltage amplifier (DLPVA, Femto Messtechnik GmbH, Berlin, Germany) and recorded using the analog input (ME 2100, Multi Channel Systems MCS GmnH, Germany). Stimulation currents were measured

across an ex vivo retina of a *rd10* mouse (female; post-natal day 114; strain: *Pde6b^{rd10}* JAX Stock No: 004297).

We applied sinusoidal voltages of 25 and 40 Hz. For 25 Hz, we applied amplitudes from 100 to 600 mV with steps of 100 mV. For 40 Hz all amplitudes were halved.

## Procedure to infer the electrical parameters of the retina

We estimated the conductivity $\sigma_{retina}$ and relative permittivity $\varepsilon_{retina}$ of the retina in three steps based on the experimental voltages $v_{stim}$ and the respective recorded currents $i_{retina}^{rec}$ and $i_{ames}^{rec}$. To facilitate the following steps, we fitted sinusoids $i_{retina}$ and $i_{ames}$ to the slightly skewed recorded currents and used them in the following (Figure 8C). To fit the sinusoids, we minimized the mean squared error between recorded currents and idealized sinusoidal currents of the same frequency $f$, resulting in estimates of the phase $\phi(i_{ames})$ and the amplitude $A(i_{ames})$ of the currents:

$$\phi(x), A(x) = \underset{\phi, A}{\operatorname{argmin}} \int_t (x - A \cdot \sin(2\pi f t + \phi))^2 dt. \tag{19}$$

During parameter inference, we only used two voltage amplitudes per frequency, resulting in four voltage and eight current traces. The other amplitudes were used for model validation. First, we estimated the electrical properties of the electrode. Here, 'electrode' is meant to include the electrical double layer and all parasitic resistances and capacitances in the electrical circuit. We simulated the voltage $v_{ames}$ across the electrolyte without retinal tissue by applying the currents $i_{ames}$ as stimuli (Figure 8Ai). Since this setup does not contain anything besides the electrolyte and the electrode, the difference between the experimental stimulus $v_{stim}$, which was applied to record $i_{ames}$, and the simulated voltage $v_{ames}$ was assumed to have dropped over the electrode:

$$v_{electrode} = v_{stim} - v_{ames}.$$

Based on that assumption, we could estimate the electrical properties of the electrode. We modeled the electrode as a *RC* parallel circuit (Figure 8Aii). Having both, sinusoidal voltages ($v_{electrode}$) over and the respective sinusoidal currents ($i_{ames}$) through the electrode, we analytically computed the values for $R_e$ and $C_e$ as follows. We assumed $R_e$ and $C_e$ to be dependent on $v_{stim}$ and therefore to be dependent on the stimulus frequency and amplitude. From the data we derived the phase $\phi_Z$ and amplitude $|Z|$ of the impedance formed by the *RC* circuit. For every $v_{electrode}$, we estimated $\phi(v_{electrode})$ and $A(v_{electrode})$ using **Equation 19**. $\phi_Z$ and $|Z|$ were then computed as:

$$\phi_Z = \phi(v_{electrode}) - \phi(i_{ames}), \qquad |Z| = A(v_{electrode})/A(i_{ames}). \tag{20}$$

Then, knowing the frequency $f$, $\phi_Z$ and $|Z|$ are sufficient to compute $R_e$ and $C_e$:

$$R_e = |Z|\sqrt{1 + tan(\phi_Z)^2}, \qquad C_e = -tan(\phi_Z)/(2\pi f R_e). \tag{21}$$

With the estimated values of the *RC* circuit, we created a model with only two unknowns, the conductivity $\sigma_{retina}$ and the relative permittivity $\varepsilon_{retina}$ of the retina (Figure 8Aiii). To estimate the unknown parameters of this model, we used the same inference algorithm as for the neuron models but with a different discrepancy function. Here, the discrepancy $\delta_R(v_{stim})$ for a stimulus $v_{stim}$ was computed as the mean squared error between the respective experimental current (now with retinal tissue) $i_{retina}$ and the simulated current $i_{retina}^{sim}$:

$$\delta_R(v_{stim}) = \sum_{v_{stim}} \int_t (i_{retina} - i_{retina}^{sim})^2 dt. \tag{22}$$

The total discrepancy was computed as the sum of all discrepancies $\delta_R(v_{stim})$ for the four different $v_{stim}$ stimuli that were used. To cover a wider range of possible parameters, we first estimated the parameters in a logarithmic space by sampling the exponents $p_\sigma$ and $p_\varepsilon$ of the parameters:

$$\sigma_{retina} = 2^{p_\sigma} \cdot 0.1 \, \text{S/m}, \qquad \epsilon_{retina} = 2^{p_\varepsilon} \cdot 10^6. \tag{23}$$

We used normal distributions (without truncation) as priors for $p_\sigma$ and $p_\varepsilon$ and set the means to 1.0 and the standard deviations to 2.0. After three rounds with 50 samples each, we computed the

minimum ($a_\sigma$, $a_\varepsilon$), maximum ($b_\sigma$, $b_\varepsilon$) and mean ($\mu_\sigma$, $\mu_\varepsilon$) for both parameters $\sigma_{retina}$ and $\varepsilon_{retina}$ from the 10% best samples. Then, we then ran the parameter inference algorithm again, but now in a linear parameter space around the best samples observed in the logarithmic space. For the priors of $\sigma_{retina}$ and $\varepsilon_{retina}$, we used truncated normal priors bound to $[a_\sigma, b_\sigma]$ and $[a_\varepsilon, b_\varepsilon]$ with means $\mu_\sigma$ and $\mu_\varepsilon$, respectively. As for the cell parameter inference, we normalized the parameter space to values between in [0, 1]. The diagonal entries of the prior covariance matrix were set to $0.3^2$, with non-diagonal entries of zero. The parameters resulting in the lowest sampled discrepancy during optimization are referred to as the optimized parameters and were used to simulate the neural responses to electrical stimulation.

## Simulation of the neural response to electrical stimulation

With the optimized parameters for the electrical properties of the retina, we were able to compute the BC responses for any given stimulation current. Note that for this, we used the model illustrated in *Figure 3* as described earlier but with the optimized parameters for $\sigma_{retina}$ and $\varepsilon_{retina}$. To simulate the neural response, we first used the stimulation current to simulate the extracellular voltage over time within the retina. After defining the relative position of the multicompartment model with respect to the retinal cylinder, we extracted the extracellular voltage for each compartment at its the central position (*Figure 3C*). Finally, these extracellular voltages were applied to the compartment models in NeuronC to simulate their response (*Figure 1C*). To estimate the uncertainty of the BC responses to electrical stimulation, we simulated different cell parametrizations in every stimulation setting. For this, we used the five best posterior samples, that is, the five (out of 200) samples with the smallest $\delta_{tot}$, for both BC models. In all simulations, we modeled subretinal stimulation of photoreceptor degenerated retina (*Zrenner, 2002*). For this, we removed all cone input from the BCs and virtually placed the multicompartment models in the retinal cylinder such that the dendrites were facing towards the electrode. The z-position of BC somata, that is, the distance to the bottom of the retinal cylinder, was set to 30 μm.

## Model validation

To validate the model for electrical stimulation, we compared simulated BC responses to experimentally recorded retinal ganglion cell (RGC) thresholds to 4 ms biphasic current pulses reported in *Corna et al., 2018*. In this study, the RGC thresholds were recorded epiretinally under subretinal stimulation of photoreceptor degenerated (*rd10*) mouse retina using a micro-electrode array (*Figure 3A*). The stimulation threshold was defined as the charge delivered during the anodic stimulation phase evoking 50% of the firing rate of a specific RGC. On the micro-electrode array. The 30 μm diameter electrodes were arranged on a regular grid with a center-center spacing of 70 μm. The RGC thresholds were measured for different numbers $N$ of $N{\times}N$ active electrodes.

We simulated the electrical field in the retina for the configurations with $1{\times}1$, $2{\times}2$, $4{\times}4$ and $10{\times}10$ active electrodes using the respective currents from the experimental data. The electrodes were centered with respect to the retina. For every stimulation current, we simulated the response of the OFF- and ON-BC at six xy-positions with distances from 0 to 500 μm relative to the center. Simulation temperature $T_{sim}$ was set to 33.5°C to match experimental conditions. For every 40 ms simulation, we computed the mean number of vesicles released per synapse.

## Optimizing electrical stimulation to separately activate ON- and OFF-BCs

To find stimuli for selective stimulation of ON- and OFF-BCs, we simulated the response of the BC models to different electrical stimuli. For this, we used a single 30 μm diameter electrode and centered the dendrites of the simulated BCs above this electrode. To find stimuli that stimulate the OFF-BC without stimulating the ON-BC or vice versa, we utilized the same algorithm used for estimating the BC parameters. Here, the inference algorithm was used to estimate parameters of a 40 ms stimulation current $i_{stim}$ parametrized by four free parameters $p_1, ..., p_4$. The current was defined as a cubic spline fit through the knot vector $\mathbf{a} = (0, p_1, ..., p_4, p^*, 0)$ spaced equidistantly in time between zero and 40 ms, where $p^*$ is chosen such that the stimulus is charge neutral (i.e. the integral over the current is zero). For all stimuli, the maximum stimulus amplitude was normalized to 0.5 μA. An illustration is shown in Figure 10.

Here, the priors over $p_1, ..., p_4$ were Gaussian with zero means and standard deviations of 0.3. For every sampled stimulus $i^n_{stim}$, we simulated the response of the BCs for $\Delta t = 60\,\mathrm{ms}$ starting with the stimulus onset. For parameter estimation, we defined the discrepancy measure $\delta^t_{stim}$ as the ratio between the relative release $R_n$ of the OFF- and ON-BC which was defined as:

$$R_n = \frac{(\mu(\mathbf{r}_n) - \mu(\mathbf{r}_{base}))\Delta t}{v^{max}_{RRP} + (r_{msr} - \mu(\mathbf{r}_{base}))\Delta t}, \tag{24}$$

where $\mu(\mathbf{r}_n)$ is the evoked mean release and $\mu(\mathbf{r}_{base})$ is the base release rate in the absence of electrical stimulation; that is, the numerator is equal to the number of released vesicles (as a mean over all synapses) caused by the stimulation. The denominator is equal to the theoretical maximum of releasable vesicles per synapse (see *Equation 9*). $\delta^t_{stim}$ was computed as:

$$\delta^t_{stim}(i^n_{stim}) = R_n(\text{other BC}) \,/\, R_n(\text{target BC}). \tag{25}$$

We ran the parameter inference twice (each with one round only), once with the ON- and once with the OFF-BC as target. We drew 400 samples from the prior that were reused for the second run of inference, and 100 more samples from the posterior. Here, the posteriors were two-component mixture of Gaussians without truncation.

## Code and data availability

Models and simulation code is available at https://github.com/berenslab/CBC_inference (*Oesterle, 2020*; copy archived at swh:1:rev:2b8ec4ac0ca916d42c-ba0404229298f8ff79c3a3). Experimental and inference data is available at https://zenodo.org/record/4185955.

## Results

We used a high-resolution electronmicroscopy data set (*Helmstaedter et al., 2013*) to create biophysically realistic multicompartment models of three neuron types from the mouse retina including cones, an OFF- and an ON-bipolar cell (BC) type. These neurons form the very beginning of the visual pathway, with cones converting light into electrochemical signals and providing input via sign-preserving and -reversing synapses to OFF- and ON-BCs, respectively. The parameters of these models include the basic electrical properties of the cells as well as the density of different ion channel types in different compartments. Given a set of parameters, simulations from the model can easily be generated; however, it is not possible to evaluate the likelihood for a given set of parameters, which would be required for standard Bayesian inference procedures for example through MCMC.

To overcome the challenge of choosing the resulting 20 to 32 parameters of these models, we adapted a recently developed technique called Sequential Neural Posterior Estimation (SNPE) (*Lueckmann et al., 2017*) (for details, see Materials and methods). Starting from prior assumptions about the parameters, the algorithm compared the result of simulations from the model to data obtained by two-photon imaging of the glutamate release from the neurons (*Franke et al., 2017*) and measured a discrepancy value between the simulation and the data. Based on this information, the algorithm used a neural network to iteratively find a distribution over parameters consistent with the measured data. This yielded optimized biophysically realistic models for the considered neuron types.

## Inference of cone parameters

We first estimated the posterior distribution over the parameters of a cone based on the glutamate release of a cone stimulated with a full-field chirp light stimulus, consisting of changes in brightness, temporal frequency, and temporal contrast (*Figure 4A* and *Figure 5*). The cone model had a simplified morphology and consisted of four compartments (*Figure 1*, see Materials and methods). We included a number of ion channels in the model reported to exist in the cones of mice or closely related species (see *Table 1*). Prior distributions were chosen based on the literature. For inference, we drew 2000 samples of different parameter settings per round and stopped the algorithm after the fourth round. Then, 200 more parameter samples were drawn from the respective posteriors for further analysis. The chosen discrepancy functions penalized discrepancies between the target and

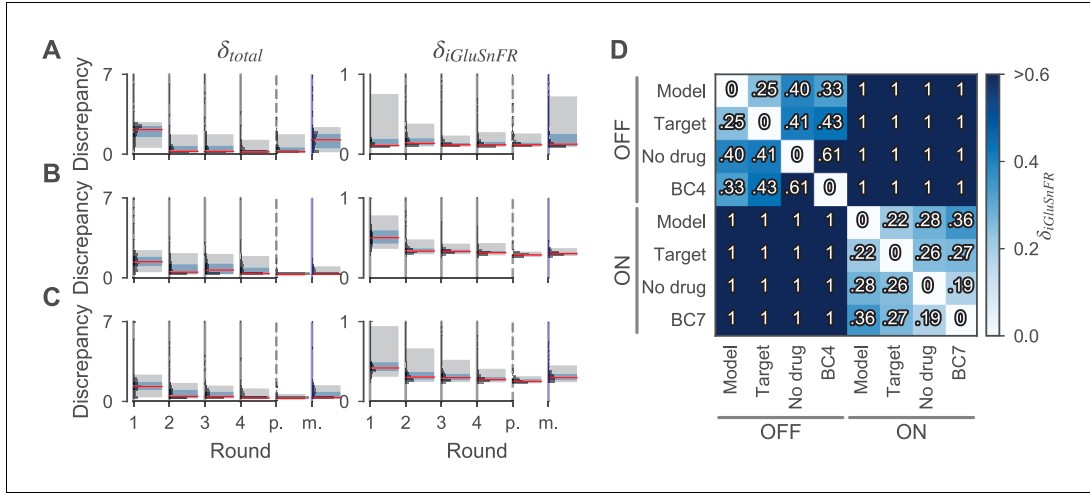

**Figure 4.** Discrepancies of samples from the cone and the BC models during and after parameter estimation. (A, B, C) Sampled discrepancies for the cone (A), the OFF- (B), and ON-BC (C) respectively. For every model, the total discrepancy $\delta_{tot}$ (left) and the discrepancy between the simulated and target iGluSnFR trace $\delta_{iGluSnFR}$ (right) are shown. For every model, four optimization rounds with 2000 samples each were drawn (indicated by gray vertical lines). After the last round (indicated by dashed vertical lines, 'p.'), 200 more samples were drawn from the posteriors. For the BCs, the number of compartments was increased in this last step to 139 and 152 for the OFF- and ON-BC, respectively. Additionally, 200 samples were drawn from assuming independent posterior marginals for comparison (indicated by blue vertical lines, 'm.'). For every round, the discrepancy distribution (horizontal histograms), the median discrepancies (red vertical lines), the 25th to 75th percentile (blue shaded area) and the 5th to 95th percentile (gray-shaded area) are shown. (D) Discrepancies between different iGluSnFR traces of BCs to demonstrate the high precision of the model fit. The pairwise discrepancy computed with equation *Equation 12* between eight iGluSnFR traces is depicted in a heat map. The column and row labels indicate which $\nu_t$ and $\nu_m$ were used in equation *Equation 12* respectively. The traces consists of the optimized BC models ('Model'), the targets used during optimization ('Target'), experimental data from the same cell type without the application of any drug ('No drug') and experimental data from another retinal CBC type with the application of strychnine ('BC4' and 'BC7'). Note that strychnine was also applied to record the targets.

The online version of this article includes the following source data for figure 4:

**Source data 1.** Sample discrepancies of all samples shown in (A-C).
**Source data 2.** Discrepancies shown in (D), and the respective (mean) iGluSnFR traces.

---

simulated iGluSnFR trace $\delta_{iGluSnFR}$, implausible membrane potentials, and implausible release rates. To compare different model fits, the discrepancy measures were added to yield a total discrepancy $\delta_{tot}$. We found that the total discrepancy $\delta_{tot}$ of the cone model was relatively high for most samples drawn from the prior but decreased over four rounds of sampling (*Figure 4A*). The discrepancy measuring the fit quality to the glutamate recording $\delta_{iGluSnFR}$ was already relatively small in the first round for most, but not all samples. In the following rounds, the number of samples with large $\delta_{iGluSnFR}$ was strongly reduced (*Figure 4A*).

The parameter setting with lowest discrepancy ($\delta_{tot} = 0.10$) modeled accurately the response of the cone to full-field stimulation with the chirp light stimulus (*Figure 5*). The simulated iGluSnFR signal nicely matched the data both on a coarse timescale and in the millisecond regime (*Figure 5D*). Indeed, for this sample, all discrepancies besides $\delta_{iGluSnFR}$ were zero or almost zero ($\delta_{tot} - \delta_{iGluSnFR} < 0.0001$) and most of the remaining discrepancy could be attributed to the noisy target data.

As our inference algorithm returned not only a single best set of parameters, but also a posterior distribution, we could obtain additional parameter samples from the model which should produce simulations consistent with the data. Almost all samples from the posterior yielded simulations that matched the target data well (median $\delta_{iGluSnFR}$: 0.12) and the overall total discrepancy was small (median $\delta_{tot}$: 0.21). Besides the discrepancy between the experimental and simulated glutamate trace $\delta_{iGluSnFR}$, most of the remaining discrepancy in the posterior samples was caused by rate

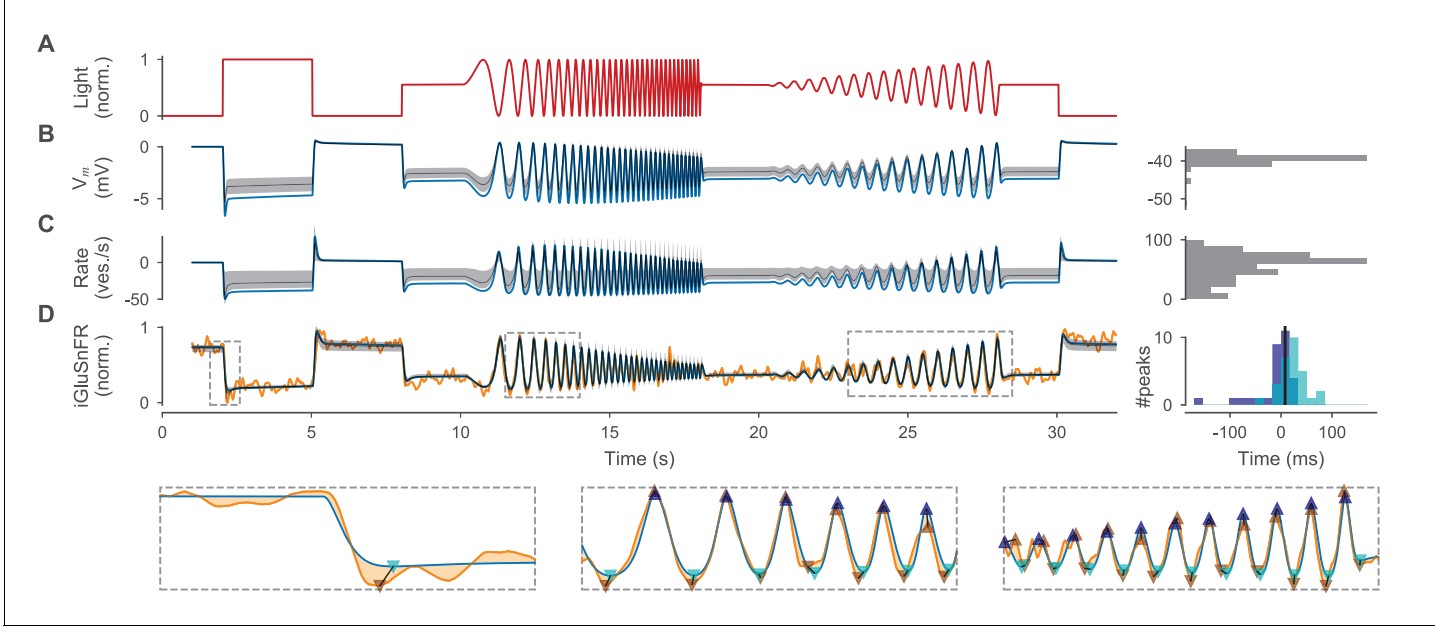

**Figure 5.** Optimized cone model. (**A**) Normalized light stimulus. (**B**) Somatic membrane potential relative to the resting potential for the best parameters (blue line) and for 200 samples from the posterior shown as the median (gray dashed line) and 10th to 90th percentile (gray shaded area). A histogram over all resting potentials is shown on the right. (**C**) Release rate relative to the resting rate. Otherwise as in (**B**). (**D**) Simulated iGluSnFR trace (as in (**B**)) compared to target trace (orange). Three regions (indicated by gray dashed boxes) are shown in more detail below without samples from the posterior. Estimates of positive and negative peaks are highlighted (up- and downward facing triangles respectively) in the target (brown) and in the simulated trace (blue and cyan respectively). Pairwise time differences between target and simulated peaks (indicated by triangle pairs connected by a black line) are shown as histograms for positive (blue) and negative (cyan) peaks on the right. The median over all peak time differences is shown as a black vertical line.

The online version of this article includes the following source data and figure supplement(s) for figure 5:

**Source data 1.** iGluSnFR traces used for constraining the cone and BC models.
**Source data 2.** Stimulus, target, and cell responses, including responses with removed ion channels.
**Figure supplement 1.** iGluSnFR traces used for constraining the cone and BC models.
**Figure supplement 2.** Effect of active ion conductances on the optimized cone model.

variation (mean $\|\delta_{Rate}^{\Delta}\|$: 0.18) and resting rates (mean $\|\delta_{Rate}^{Rest}\|$: 0.14) that were too low in some of the models. While in principle we could propagate the remaining uncertainty about the model parameters provided by the posterior to the inference about BC models, we used only the parameter set with the smallest total discrepancy $\delta_{tot}$ for efficiency and refer to this as the optimized cone model. To analyze the role of active ion channels, we removed ion channels individually (except for the $Ca_L$ channel with is necessary to simulate the vesicle release) from the optimized cone and simulated the light response (***Figure 5—figure supplement 2***). We found that the HCN channel contributed most, while the contribution of $Cl_{Ca}$ was negligible. Since $Cl_{Ca}$ did not alter the light response for both, the cone and BC light stimulus, we removed it in the following steps for computational efficiency.

## Inference of bipolar cell parameters

We next turned to anatomically detailed multicompartment models of two BC types. We chose to model type 3a and type 5o because these were the OFF- and ON-CBC types for which we could gather most prior information from the literature. The anatomy of the cells was extracted from a 3D reconstruction of examples of these cell types based on electron microscopy data (***Helmstaedter et al., 2013***) and divided into regions sharing ion channel parameters (***Figure 1***). As for the cone model, the channels included in the model and the prior distributions were chosen based on the literature (see ***Table 1***). This yielded 32 and 27 free parameters for the OFF- and ON-BC, respectively.

We fitted the BC type models to published two-photon glutamate imaging data reflecting the glutamate release from the BC axon terminals (***Franke et al., 2017***). In this case, we used responses

to a local chirp light stimulus activating largely the excitatory center of the cells. In addition, the responses were measured in the presence of the drug strychnine to block locally acting inhibitory feedback through small-field amacrine cells (*Franke et al., 2017*) (see Materials and methods for details). Similar to what we observed in cones, the total discrepancy $\delta_{tot}$ for parameter sets sampled for the OFF- and ON-BC model decreased over the four rounds of optimization (*Figure 4B and C*). In contrast to the the cone model, the discrepancy measure penalizing deviations from the gluta-mate trace $\delta_{iGluSnFR}$ was relatively large for prior samples and declined approximately equally fast as the total discrepancy $\delta_{tot}$.

We found that simulations generated with the parameter set with minimal total discrepancy or parameters sampled from the posterior matched the target traces very well for both OFF- and ON-BC models (*Figure 6*). For these parameters, the cells were relatively isopotential units throughout the light stimulus (*Figure 6—figure supplement 1* and *Figure 6—figure supplement 2*) with a larger voltage gradients from dendrites to the axon in the ON-BC. The optimized BC models, that is the BC samples with the lowest total discrepancies $\delta_{tot}$, had discrepancies of zero except for the iGluSnFR discrepancy $\delta_{iGluSnFR}$. To get a more quantitative impression of the quality of the model fits, we compared the pairwise iGluSnFR discrepancies $\delta_{iGluSnFR}$ between the optimized BC models, the experimentally measured response traces as used during optimization, traces recorded from the same cell type without application of strychnine and responses of another OFF- and ON-BC. For both optimized cell model outputs, the discrepancy was smallest for the targets used during optimi-zation. This shows that the optimized models were able to reproduce cell-type specific details in light response properties that go beyond the simple distinction of ON and OFF responses. While the discrepancies between traces of different ON-BC types were overall relatively small for local stimulation (*Franke et al., 2017*), the discrepancies between traces from OFF cells were larger likely due to network modulation of the target cell type by amacrine cells (indicated by the difference between the target and the no-drug condition) and larger response differences between the two compared OFF-BC types. The posterior samples of both BC models had a low discrepancy, except for a few samples (median $\delta_{tot}$: 0.29 and 0.26 of the OFF- and ON-BC, respectively). The only dis-crepancy measure with a non-zero median of the absolute values was $\delta_{iGluSnFR}$, which also accounts for 88% and 82% of the mean total discrepancy for the OFF- and ON-BC respectively.

Despite the overall high resemblance between optimized model outputs and targets, there were some visible systematic differences. For the ON-BC, the target showed a skewed sinusoidal response with faster rise than fall times during the frequency and amplitude increasing sinusoidal light stimulation between 10 s and 18 s and between 20 s and 27 s respectively. In contrast, the opti-mized model output showed approximately equal rise and fall times, resulting in a systematic delay of positive and negative peaks (median delay of all peaks: 15.6 ms) in the simulated iGluSnFR trace relative to the target (*Figure 6G*). Additionally, some of the positive peaks of the optimized ON-BC model during sinusoidal light stimulation were too small (e.g. at 11.5 s). This effect might have been a side-effect of the peak timing difference between target and model: Amplitude increases were inefficient in reducing the discrepancy as long as the peaks were not precisely aligned. In contrast, the peak time precision of the OFF-BC model (*Figure 6D*) was much higher (median delay of all peaks: 0.0 ms). In this case, the main difficulty for the model appeared to be its inability to repro-duce the non-linearity in the cell response to the increasing amplitude sinusoidal light stimulation between 20 s and 27 s.

After having verified that the posterior over parameters provided a good fit to the experimental data, we inspected the one-dimensional marginal distributions to learn more about the resulting cel-lular models (*Figure 7*). For most parameters, the marginal posteriors had smaller variances than the priors, indicating that the parameter bounds were not chosen too narrow. For some parameters, the posterior mean differed substantially from the prior mean (e.g. the $Ca_T$ channel density at the axon terminal of OFF-BC) while it was largely unchanged for others (e.g. the $Ca_L$ channel density at the soma for the OFF-BC). The algorithm also inferred the dependencies of some parameters, visible in the two-dimensional marginals (*Figure 7—figure supplement 1* and *Figure 7—figure supplement 2*). Because of these correlations, the full posterior in the high-dimensional parameter space led to simulations which were on average better (median: 0.29 vs. 0.31 and 0.26 vs. 0.33 for the OFF- and ON-BC, respectively) and less variable in their quality (95%-CIs: 0.53 vs. 1.01 and 0.64 vs. 1.42 for the OFF- and ON-BC, respectively) than parameters drawn from a posterior obtained by assuming independent marginal distributions. In most cases, the parameters resulting in the lowest total

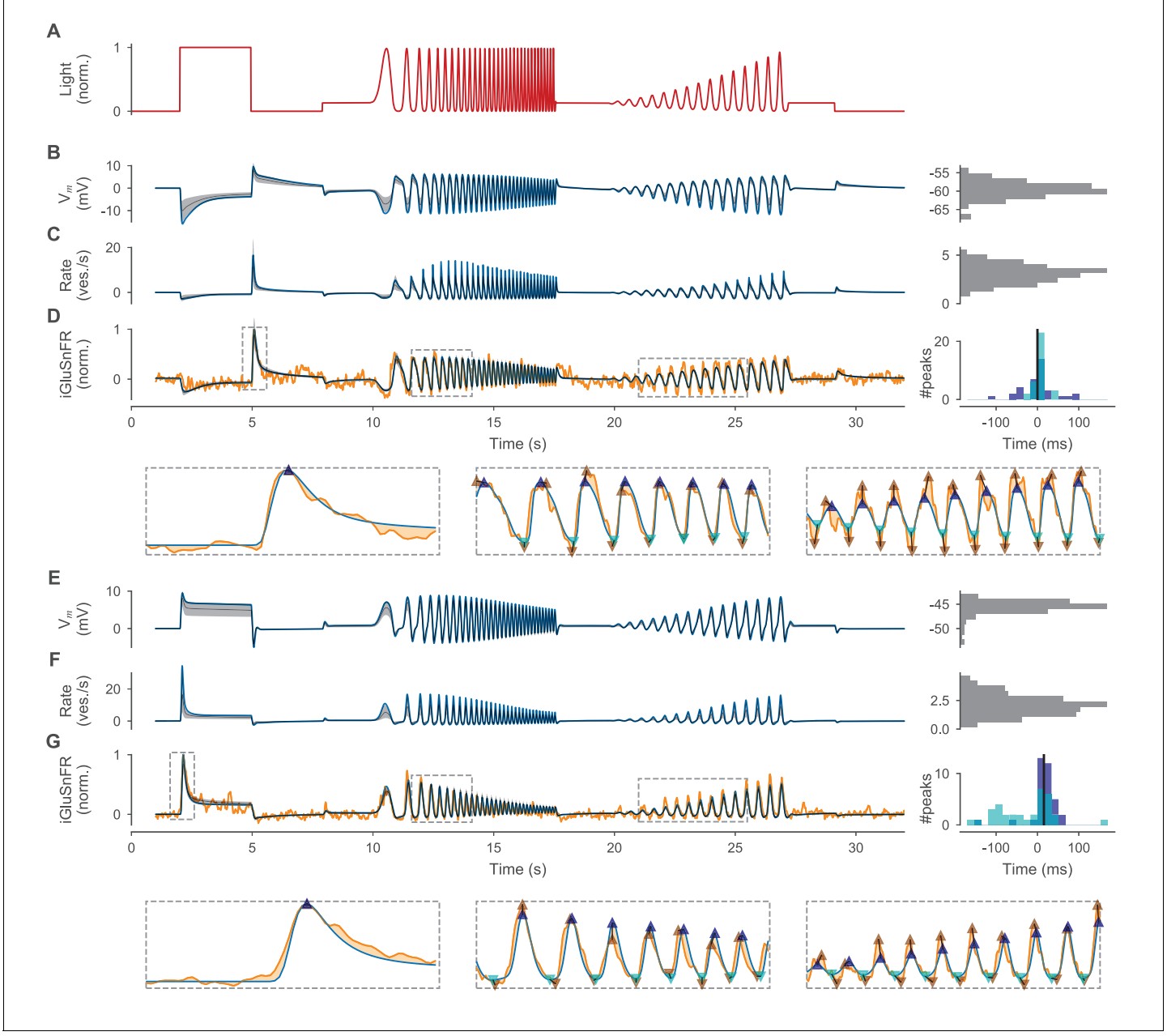

**Figure 6.** Optimized BC models. (**A**) Normalized light stimulus. Responses of the OFF- and ON-BC are shown in (**B–D**) and (**E–G**), respectively. (**B, E**) Somatic membrane potential relative to the resting potential for the best parameters (blue line) and for 200 samples from the posterior shown as the median (gray dashed line) and 10th to 90th percentile (gray-shaded area). A histogram over all resting potentials is shown on the right. (**C, F**) Mean release rate over all synapses relative to the mean resting rate. Otherwise as in (**B**). (**D, G**) Simulated iGluSnFR trace (as in (**B**)) compared to respective target trace (orange). Three regions (indicated by gray dashed boxes) are shown in more detail below without samples from the posterior. Estimates of positive and negative peaks are highlighted (up- and downward facing triangles, respectively) in the target (brown) and in the simulated trace (blue and cyan, respectively). Pairwise time differences between target and simulated peaks (indicated by triangle pairs connected by a black line) are shown as histograms for positive (blue) and negative (cyan) peaks on the right. The median over all peak time differences is shown as a black vertical line. The online version of this article includes the following video, source data, and figure supplement(s) for figure 6:

**Source data 1.** Stimulus, target, and cell responses, including responses with removed ion channels.

**Figure supplement 1.** Heatmaps of the OFF-BC.

**Figure supplement 2.** Heatmaps of the ON-BC.

**Figure supplement 3.** Effect of active ion conductances on the optimized OFF-BC model.

**Figure supplement 4.** Effect of active ion conductances on the optimized ON-BC model.

**Figure 6—video 1.** Animated heatmaps of the OFF-BC.

*Figure 6 continued on next page*

*Figure 6 continued*

**Figure 6—video 2.** Animated heatmaps of the ON-BC.

discrepancy were close to the means of the respective posteriors. For some parameters there was a strong difference between the marginal posteriors of the OFF- and ON-BC. For example, the two parameters controlling the leak conductance, $V_r$ and $R_m$, were much lower for the OFF-BC consistent with the strong variation of membrane resistances reported in *Oltedal et al., 2009*. The membrane conductance was lower for the ON-BC, which could increase signal transduction speed in the longer axon. Even though the posteriors were narrower than the priors, they still covered a wide range of different parameters. To some extent, this may reflect the fact that we fit the model parameters solely on the cells output, and for example dendritic parameters may be underconstrained by such data; in addition, it may also reflect variability between cells of the same type seen in the experimental data that has also been reported in other studies (*Franke et al., 2017*).

After the fourth optimization round, 200 samples were drawn from the posterior distribution with an increased number of compartments to find model parameters to simulate electrical stimulation (see Methods). For comparison, we also ran simulations with the same parameters but the original number of compartments (data not shown). Interestingly, more than 85% of the samples had a lower discrepancy if the models were simulated with more compartments for both BCs. For the best 20% (i.e. 40 samples) of the posterior samples (sorted with respect to samples with fewer compartments), the samples with more compartments had lower discrepancies with only one exception per cell. This

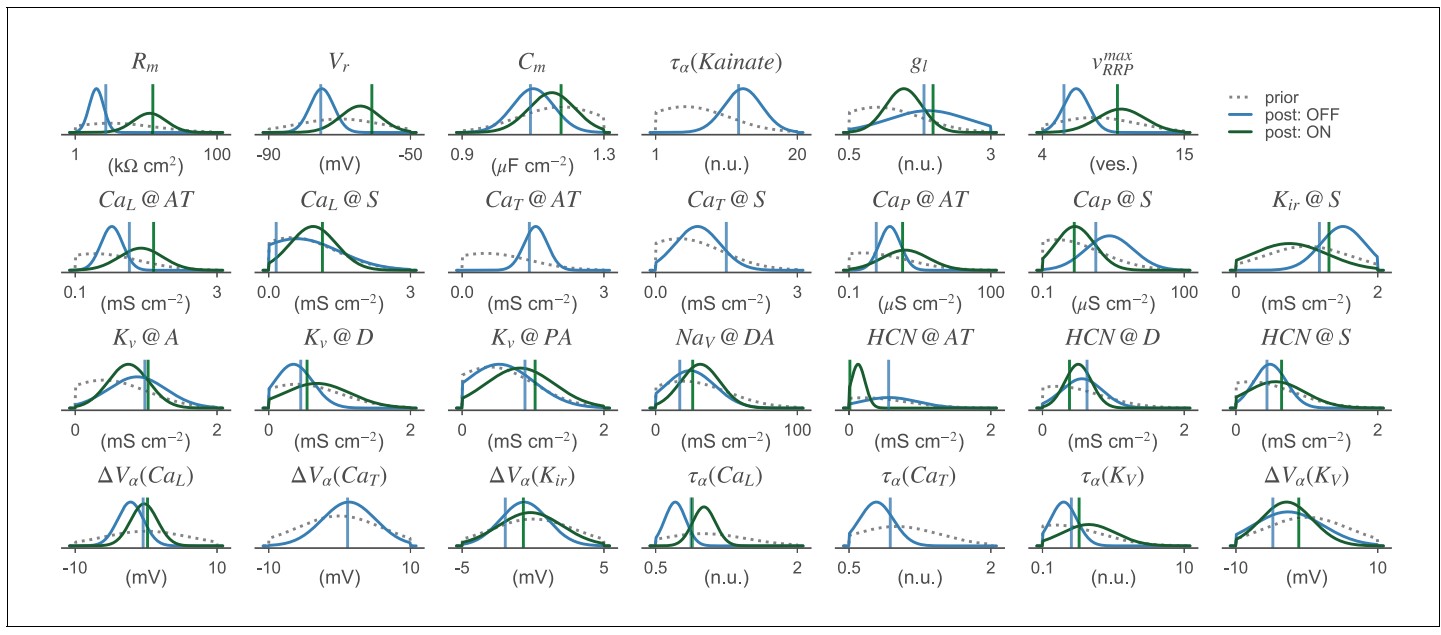

**Figure 7.** Parameter distributions of the BC models. 1D-marginal prior (dashed gray line) and posterior distributions (solid lines) are shown for the OFF- (blue) and ON-BC (green). The parameters of the posterior samples with the lowest total discrepancy are shown as dashed vertical lines in the respective color. $X_Y@Z$ refers to the channel density of channel $X_Y$ at location $Z$. Locations are abbreviated; S: soma, A: axon, D: dendrite and AT: axon terminals (see *Figure 1* and main text for details). Note that although these 1D-marginal distributions seem relatively wide in some cases, the full high-dimensional posterior has much more structure than a posterior distribution obtained from assuming independent marginals (see *Figure 4*). Not all parameter distributions are shown.

The online version of this article includes the following source data and figure supplement(s) for figure 7:

**Source data 1.** Prior and posterior parameters for the OFF- and ON-BC.
**Figure supplement 1.** 2D-marginals for the OFF-BC.
**Figure supplement 2.** 2D-marginals for the ON-BC.

indicates that, given enough computational power, the same parameter inference approach but with more compartments could further improve the model outputs. From these samples, we used the five samples with the smallest total discrepancies $\delta_{tot}$ for the simulation of electrical stimulation.

Additionally, we used these five samples to analyze the effect of active ion conductances on the light response by removing individual ion channels types from the BCs (*Figure 6—figure supplement 3* and *Figure 6—figure supplement 4*). Similar to the optimized cone model, the HCN channels played the most important role in shaping the light response. For both cells, the $\mathrm{Na_V}$ and somatic calcium channels barely had any influence on the membrane voltage or the vesicle release rate.

## Simulating electrical stimulation of the retina

To provide an exemplary use-case for our data-driven biophysical models of retinal neurons, we asked whether we could use our simulation framework to optimize the stimulation strategy for retinal neuroprosthetic implants. These implants have been developed for patients who lose sight due to degeneration of their photoreceptors (*Zrenner, 2002*). While existing implants have been reported to provide basic vision (*Zrenner, 2002*; *Edwards et al., 2018*; *Luo and da Cruz, 2016*), they are far from perfect. For example, most current stimulation strategies likely activate OFF- and ON-pathways at the same time (*Barriga-Rivera et al., 2017*). To this end, we created a simulation framework for subretinal electrical stimulation of retinal tissue with micro-electrode arrays. We estimated the conductivity and relative permittivity of the retina based on experimentally measured currents evoked by sinusoidal voltages and then validated simulations of the electrical stimulation of our fitted BC models with standard pulse like stimuli against responses measured in RGCs (*Corna et al., 2018*). Finally, we used the simulation framework to find stimuli that can separately stimulate OFF- and ON-BCs.

Our framework for simulating the effect of external electrical stimulation using the inferred BC models consisted of two steps: we first estimated the electrical field resulting from a stimulation protocol as a function of space and time across the whole retina (*Figure 3*). The corresponding extracellular voltages were then applied to the respective compartments to simulate the neural response. To do so, we needed a model of the electrical properties of the electrode and the retinal tissue. We assumed disk electrodes and a simplified model assuming homogeneous electrical properties of the retina and the surrounding medium (see Materials and methods). This model contained two free parameters that needed to be estimated from data: the conductivity and relative permittivity of the retinal tissue.

To estimate these parameters we recorded electrical currents resulting from sinusoidal voltage stimulation with different frequencies in a dish with and without a retina (*Figure 8B,C*). We used the data without a retina to estimate the properties of the stimulation electrode (*Figure 8A,D,E* and Materials and methods). Based on the estimates of the electrode properties and the data recorded with a retina, we estimated the conductivity and relative permittivity of the retina with the same parameter inference method as for the neuron models.

We found that both parameters are very well constrained by the measured data (*Figure 8F*). The parameters resulting in the lowest discrepancy were $\sigma_{retina} = 0.076\,\mathrm{S/m}$ and $\varepsilon_{retina} = 1.1 \times 10^7$ in accordance with the conductivity of $0.077\,\mathrm{S/m}$ reported for rabbit *Eickenscheidt et al., 2012* and the relative permittivity of gray matter estimated in *Gabriel et al., 1996*. With these parameters, we simulated currents for all stimulus amplitudes we recorded experimentally. The simulated and experimental currents matched for the amplitudes used during parameter inference but also for amplitudes reserved for model validation (*Figure 8G*). Therefore, we used them in all following experiments.

To validate our simulation framework, we compared simulated BC responses to experimentally measured RGC thresholds (*Corna et al., 2018*). We simulated BCs at different positions for four different electrode configurations (*Figure 9A*) and nine stimulation current wave forms (*Figure 9B*). For small stimulus charge densities, the BCs barely responded, whereas for very high charge densities the cells released all glutamate vesicles available in the readily release pool (*Figure 9C and D*). In between, the response increased from no response to saturation, dependent on the distances of the simulated cell to the active electrodes. Across stimulation conditions, these threshold regions coincide with the measured RGC thresholds to the same stimulation, when assuming that the stimulated RGCs were not too far away from the stimulation electrode. Since the reported RGC thresholds likely

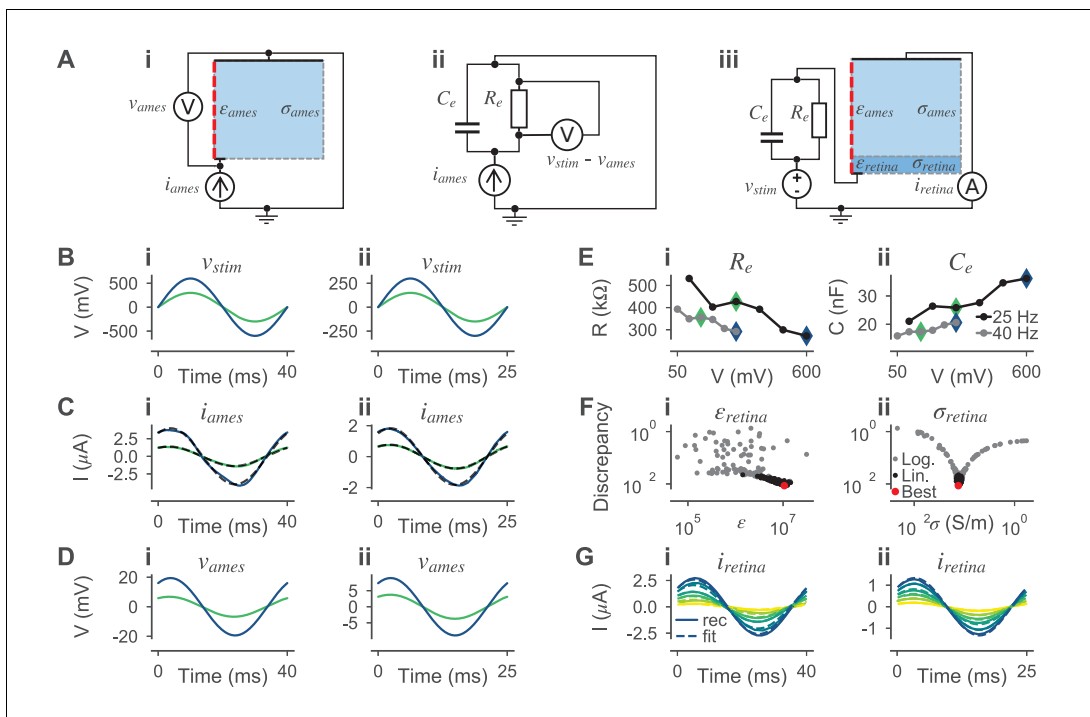

**Figure 8.** Estimation of the conductivity $\sigma_{retina}$ and the relative permittivity $\varepsilon_{retina}$ of the retina for simulating electrical stimulation. (**A**) Electrical circuits used during parameter estimation. (**B**) Stimulation voltages $v_{stim}$ at 25 (left) and 40 Hz (right). From the six experimentally applied amplitudes, only the amplitudes used for parameter inference are shown. (**C**) Experimentally measured currents $i_{ames}$ through electrolyte (Ames' solution) without retinal tissue for the stimulus voltages $v_{stim}$ in (B). The mean traces over all (but the first two) repetitions are shown (black dashed lines). Sine waves were fitted to the mean traces (solid lines) with colors referring to the voltages in (B). (**D**) Simulated voltages over the electrolyte $v_{ames}$ using the fitted currents in (C) as stimuli applied to the circuit in (Ai). (Aii) Electrical circuit used to model the electrode plus interface. (**E**) Stimulus frequency and amplitude dependent estimates of $R_e$ (i) and $C_e$ (ii) based on the electrical circuit shown in (Aii) for 25 (black) and 40 Hz (gray). Note that the values were derived analytically (see main text). The values corresponding to the stimulus voltages shown in (B) are highlighted with respective color. (Aiii) Electrical circuit used to estimate $\sigma_{retina}$ and $\varepsilon_{retina}$. The respective values for $R_e$ and $C_e$ are shown in (E) and are dependent on $v_{stim}$. The current $i_{retina}$ through the model is measured for a given stimulus voltage $v_{stim}$. (**F**) Sampled parameters of $\varepsilon_{retina}$ and $\sigma_{retina}$ and the respective sample losses. First, samples were drawn in a wide logarithmic space (gray dots) and then in a narrower linear parameter space. The best sample (lowest discrepancy) is highlighted in red. (**G**) Simulated currents $i_{retina}$ (solid lines) through the circuit in (Aiii) with optimized parameters (red dot in (F)) and respective experimentally measured currents (broken lines). Here, results for all six stimulus amplitudes are shown for both frequencies.

The online version of this article includes the following source data for figure 8:

**Source data 1.** Currents, voltages, and samples from optimization.

result from indirect stimulation via BCs, the consistency between the RGC and simulated BC thresholds can be interpreted as evidence that our model was well calibrated to simulate electrical stimulation.

## Optimized electrical stimulation for selective OFF- and ON-BC stimulation

We finally used our framework for electrical stimulation to find stimuli that excite OFF- or ON-BCs selectively. To this end, we performed Bayesian inference over an electrical charge-neutral stimulus (*Figure 10A*) with the SNPE algorithm, using the response ratio between the two BC types (*Figure 10B*) as a discrepancy function to minimize. Using this procedure, we found that triphasic, anodic first stimuli with a cathodic middle phase of high amplitude (*Figure 10C*) evoked substantial neurotransmitter release in the OFF-BC (*Figure 10Di*) while evoking almost no response in the ON-BC (*Figure 10Dii*). The stimuli optimized to target the ON-BC were biphasic, with no clear

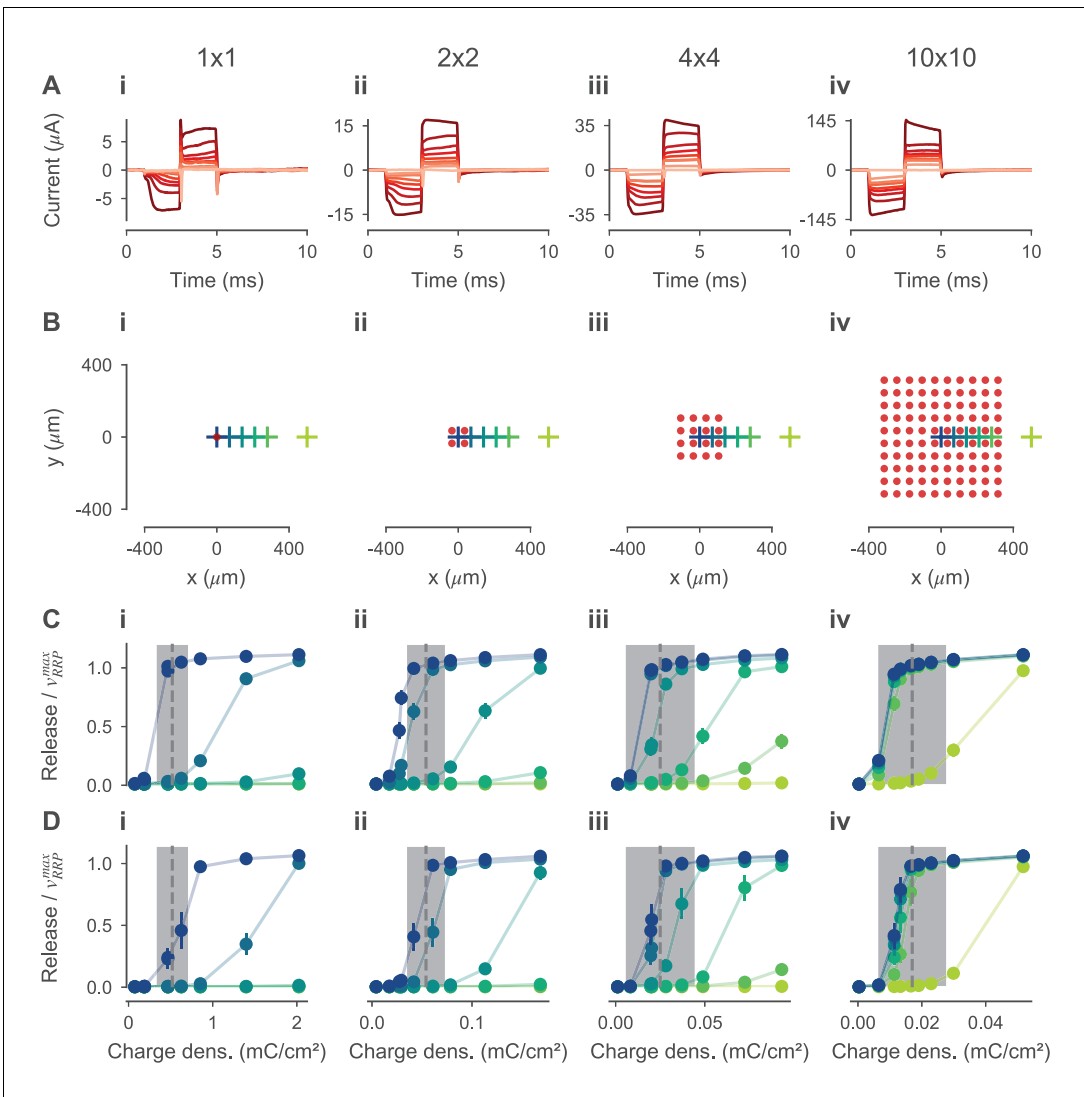

**Figure 9.** Threshold of electrical stimulation for experimentally measured RGCs and simulated BCs of photoreceptor degenerated mouse retina. (**A**) Stimulation currents measured experimentally and used as stimuli in the simulations. (**B**) xy-positions of BCs (crosses) and electrodes (red dots) for 1×1, 2×2, 4×4, 10×10 stimulation electrodes, respectively. Every electrode is modeled as a disc with a 30 μm diameter. Except for the electrode configuration, the models were as in *Figure 8*. (**C, D**) Mean synaptic glutamate release relative to the size of the readily releasable pool $v_{RRP}^{max}$ of simulated OFF- and ON-BCs, respectively. Values can be greater than 1, because the pool is replenished during simulation. Errorbars indicate the standard deviation between simulations of BCs with different parametrizations; for both BCs, the five best posterior samples were simulated. Glutamate release is shown for different charge densities (x-axis) and cell positions (colors correspond to xy-positions in (A); for example the darkest blue corresponds to the central BC). Experimentally measured RGC thresholds (gray dashed lines) plus-minus one standard deviation (gray-shaded ares) are shown in the same plots.

The online version of this article includes the following source data for figure 9:

**Source data 1.** Current traces and vesicle release for all simulations.

preference for anodic or cathodic first currents (*Figure 10E*). In contrast to the stimuli optimized for the OFF-BC, these stimuli did not exclusively stimulate the targeted ON-BC (*Figure 10Fii*) but also the OFF-BC (*Figure 10Fi*). We did not find stimuli evoking stronger release (defined as in *Equation 24*) in the ON-BC than the OFF-BC. This lower threshold of the OFF-BC, which we also observed for the biphasic current pulses (*Figure 9*), was partially caused by the shorter axon of the OFF-BC resulting in slightly larger changes of the extracellular voltage at the axon terminals during

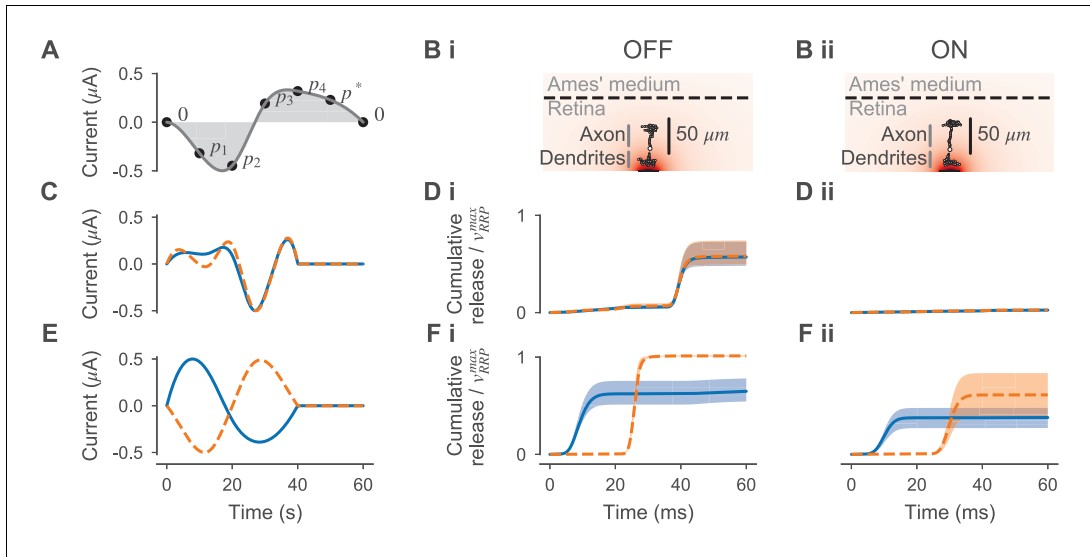

**Figure 10.** Electrical currents optimized for selective ON- and OFF-BC stimulation. (**A**) Illustration of the stimulus parametrization. A stimulation current was computed by fitting a cubic spline through points defined by $p_1...p_4$ and $p^*$ (black dots) that were placed equidistantly in time. $p_1...p_4$ were free parameters and $p^*$ was chosen such that the stimulus was charge neutral, that is the integral over the current (gray-shaded area) was zero. Currents were normalized such that the maximum amplitude was always 0.5 µA. (**B**) Illustration of the electrical stimulation of an OFF- (i) and ON-BC (ii) multicompartment model. (**C, E**) Stimuli optimized for selective OFF- and ON-BC stimulation, respectively. The two best stimuli observed during optimization are shown. (**D, F**) Cumulative vesicle release (as a mean over all synapses) relative to the size of the readily releasable pool $v_{RRP}^{max}$ in response to the electrical stimuli shown in (C) and (E), respectively. Stimuli and responses are shown in matching colors. Release is shown for the five best posterior cell parametrizations for the OFF- (i) and ON-BC (ii) as a mean (line) plus-minus one standard deviation (shaded area).

The online version of this article includes the following source data and figure supplement(s) for figure 10:

**Source data 1.** Optimized stimuli and BC responses, including responses with removed ion channels.
**Figure supplement 1.** Electrical stimulation with optimized stimuli and removed ion channels.

stimulation (*Figure 10—figure supplement 1A,B*). While the morphologies and ion channel distributions contributed to differences in the membrane voltage at the axon terminals (*Figure 10—figure supplement 1C*), the decisive factor for differences in the OFF and ON response presumably lies in the presence or absence of $Ca_T$ channels at the axon terminals. Removing these channels from the OFF-BC had almost no effect on the membrane voltage, but resulted in qualitatively similar responses for both cell types (*Figure 10—figure supplement 1D,E*). Removing the $Ca_L$ channels at the axon terminals on the other hand, left the OFF response for the triphasic and biphasic cathodic first stimuli largely unchanged (*Figure 10—figure supplement 1F*).

## Discussion

In this study, we showed how the recently developed Bayesian likelihood-free parameter inference method called Sequential Neural Posterior Estimation (SNPE) (*Lueckmann et al., 2017*) can be used to estimate the parameters of multicompartment models of retinal neurons based on light-response measurements. In addition, we built a model for electrical stimulation of the retina, and optimized electrical stimulation protocols for retinal neuroprosthetic devices designed to support vision in the blind.

Performing Bayesian inference for mechanistic models is difficult, as they typically do not come with an analytical solution of the likelihood function. The SNPE algorithm — like many approximate Bayesian computation (ABC) methods (*Sisson et al., 2018*) — does not require such an analytical formulation, as it builds on simulations of the model. In contrast to standard ABC methods, the SNPE algorithm makes efficient use of simulation time by using all simulated data to train a mixture

density network to update the parameter distributions (*Lueckmann et al., 2017*; *Gonçalves et al., 2020*). Moreover, SNPE makes minimal assumptions about the simulator, giving full flexibility to use it with different simulation environments and software. As all Bayesian methods, SNPE allows the incorporation of literature knowledge in the form of priors which can be used to constrain parameters and to put more weight on more plausible parameters. Finally, it does not only yield a point-estimate of the best parameters — like exhaustive grid-search techniques (*Goldman et al., 2001*; *Prinz et al., 2003*; *Stringer et al., 2016*) or evolutionary algorithms (*Gerken et al., 2006*; *Keren et al., 2005*; *Achard and De Schutter, 2006*; *Rossant et al., 2011*) — but also returns a posterior distribution that reflects remaining parameter uncertainties and allows one to detect dependencies between parameters.

Recently, there has been a surge of interest in Bayesian simulator-based inference with many recently published algorithms (*Gutmann and Corander, 2016*; *Papamakarios and Murray, 2016*; *Lueckmann et al., 2017*; *Lintusaari et al., 2017*; *Papamakarios et al., 2018*; *Wood, 2010*; *Durkan et al., 2018*; *Sisson et al., 2018*; *Gonçalves et al., 2020*; *Bittner et al., 2019*). While we initially evaluated different algorithms, we did not perform a systematic comparison or benchmarking effort, which is beyond the scope of this project. Much of the literature on simulator-based inference evaluates the proposed algorithms on fairly simple models. In contrast, we used SNPE here to perform parameter inference of comparatively complex multicompartment models of neurons. Importantly, we did not need targeted experiments to constrain these models, but based our framework on two-photon imaging data of glutamate release in response to simple light stimuli using a genetically encoded indicator called iGluSnFR (*Marvin et al., 2013*; *Franke et al., 2017*). This methods allows direct experimental access to the neurotransmitter release of all excitatory neurons of the retina (*Euler et al., 2019*). Using this data, we inferred the distributions of model parameters relevant for all the intermediate steps between light stimulation of cones to the glutamate release from synaptic ribbons. While we optimized many parameters in the models using SNPE, we chose to keep some parameters on sensible default values to avoid issues with computational complexity. Of course, it is possible that optimization of the full parameter space would have yielded slightly better results or that some parameters would have been set to slightly different values, as a mechanism whose parameter was allowed to vary compensated for the one whose parameter was held fixed. As an alternative to our approach, one can combine classical systems identification approaches with inference for only some of the biophysical mechanisms such as the ribbon synapse (*Schröder et al., 2019*). Our approach, however, allows the exploration of mechanisms within neurons which are not or only barely experimentally accessible. For example, in BCs, it is currently difficult to obtain direct voltage measurements from any part of the cell but the soma. If one is interested in how the electrical signal propagates through the axon or the axon terminal, our simulations may help to obtain mechanistic insights and develop causal hypotheses.

Because our inference framework works with experimental measurements which can be performed in a high-throughput manner, it allows for a comparably easy scaling to infer model parameters of a larger number of multicompartment models e.g. of different neuron types. In principle it could even be possible to infer the parameters of a neuron by imaging another neuron. For example, one could attempt to infer parameters of an amacrine cell by observing the neurotransmitter release of a connected BC — although such an indirect inference would likely result in larger uncertainties. Ideally, such a large-scale approach would also include realistic morphologies, for example from electron microscopy as shown here. In fact, BCs are anatomically relatively 'simple' neurons, and it will be interesting to test our inference methods on neurons with more complicated structure such as some amacrine cells (*Masland, 2012*). While we did not aim at an exhaustive analysis of the effect of morphology on the neuron responses, one could easily explore how details of the morphology influence the distribution of optimal biophysical parameters. Further, we extended our model to simulate and optimize external electrical stimulation of the retina. For blind patients suffering from photoreceptor degeneration, for example caused by diseases like Retinitis Pigmentosa, neuroprosthetic devices electrically stimulating remaining retinal neurons can restore basic visual abilities (*Edwards et al., 2018*; *Luo and da Cruz, 2016*). The spatial and temporal resolution of such retinal implants is, however, still very limited (*Weitz et al., 2015*) with the highest reported restored visual acuity of 20/546 (*Chuang et al., 2014*). While many experimental studies have explored different strategies of stimulation (*Jensen et al., 2005*; *Jensen and Rizzo, 2008*; *Tsai et al., 2009*; *Eickenscheidt et al., 2012*), most of them are restricted to very specific stimulus types such as

current or voltages pulses. As a consequence, retinal implants are not able to specifically target cell types such as the independent stimulation of the ON and OFF pathways of the retina (*Lee and Im, 2019*; *Barriga-Rivera et al., 2017*; *Twyford et al., 2014*). To facilitate a systematic stimulus optimization in silico, we developed a simulation framework for electrical stimulation of the retina. While the idea to simulate responses of BCs to electrical stimuli is not new, previous studies restricted their models to point/ball source electrodes (*Resatz and Rattay, 2004*; *Rattay et al., 2017*), simplified BCs to passive cables (*Gerhardt et al., 2010*) or used simplified BC models that only express L- or T-type channels (*Werginz et al., 2015*). Our framework combines the simulation of micro-electrode arrays used in neuroprosthetic devices (*Edwards et al., 2018*; *Luo and da Cruz, 2016*) with detailed models of an OFF- and ON-BC. This allowed us to test a large number of stimulus waveforms, optimizing for stimuli selectively targeting either OFF- or ON-BCs, which could help to better encode visual scenes into electrical signals of retinal implants. We found stimuli selectively targeting the OFF-BC without stimulating the ON-BC, but not vice versa. Likely, the main reason for the differential response of the two BC types was that only the OFF-BC had T-type calcium channels at the axon terminals. These channels were more sensitive to transient changes in the membrane voltage which were evoked by the stimuli optimized to selectively target the OFF-BC. The ON-BC, having no T-type calcium channels and an overall higher threshold, did not respond to these stimuli. However, it could be stimulated with longer anodic stimulus phases activating the L-type calcium channels. Since we modeled the cells in isolation, network effects through synaptic activation of amacrine cells might further modulate the activity of the BCs. However, the neurites and somata of amacrine cells are substantially farther away from the stimulation electrode than those of the BCs, and these effects might be comparably small. That notwithstanding, simulations including network effects and also more diverse BC types will be required in the future. Ideally, in silico optimized stimulation strategies would be first verified in ex vivo experiments before implementing them in neuroprosthetic devices to improve the quality of visual impressions.

To be able to simulate the electrical stimulation of the retina, we first inferred the conductivity and relative permittivity of the *rd10* retina based on recorded currents evoked by sinusoidal stimulation voltages. While the estimated conductivity ($\sigma_{\mathrm{retina}} = 0.076\,\mathrm{S/m}$) is consistent with the value ($\sigma_{\mathrm{retina}} = 0.077\,\mathrm{S/m}$) reported in *Eickenscheidt et al., 2012*, also smaller ($0.025\,\mathrm{S/m}$, *Karwoski and Xu, 1999*) and larger ($\approx 0.75\,\mathrm{S/m}$, *Wang and Weiland, 2015*) conductivities have been reported for the retina. These differences may be due to different ways in tissue handling and preparation. Comparing the estimated values of the relative permittivity ($\varepsilon_{\mathrm{retina}} = 1.1 \times 10^7$) to the literature is more difficult, and most simulation studies choose to ignore its effects. The relative permittivity of retinal tissue has been reported for very high frequencies (10 MHz to 10 GHz), but the strong frequency dependence makes a direct comparison to frequencies several orders of magnitude smaller (e.g. 40 Hz) not meaningful. Additionally, data from gray matter suggest a relative permittivity of $1.5 \times 10^7$ at 40 Hz very close to our estimate (*Gabriel et al., 1996*). This opens the question weather the common assumption to ignore the effects of the relative permittivity in other simulations (*Gerhardt et al., 2010*; *Werginz et al., 2015*; *Rattay et al., 2017*) is valid.

In summary, mechanistic models in neuroscience such as biophysically realistic multicompartment models have long been regarded as requiring many manual parameter choices or highly specific experiments to constrain them. We showed here how relatively standard, high-throughput imaging data in combination with likelihood-free inference techniques can be used to perform Bayesian inference on such models, allowing unprecedented possibilities for efficient optimization and analysis of such models. Importantly, this allow us to understand which parameters in such models are well constrained, and which are not, and determine which parameter combinations lead to similar simulation outcomes (*Gonçalves et al., 2020*; *Alonso and Marder, 2019*) — questions that have hindered progress in the field for years.

## Acknowledgements

We thank Yves Bernaerts for preliminary work on electrical stimulation of biophysical models, Timm Schubert for support with animal protocols and Gordon Eske and Adam vor Daranyi for technical support. Additionally, we thank Jakob Macke, Philipp Hennig and Lara Hoefling for discussion. This work was funded by the Federal Ministry of Education and Research (BMBF, FKZ 01GQ1601 and 01IS18052 to PB), the German Science Foundation through the Excellence Cluster 2064 'Machine

Learning - New Perspectives for Science' (390727645), a Heisenberg Professorship (BE5601/4-1 to PB), the Baden-Württemberg Stiftung (NEU013 to PB and GZ), and the National Institutes of Health (NIH) (EY022070 and EY023766 to RGS).

## Additional information

### Funding

| Funder | Grant reference number | Author |
|---|---|---|
| Bundesministerium für Bildung und Forschung | 01GQ1601 | Philipp Berens |
| Bundesministerium für Bildung und Forschung | 01IS18052 | Philipp Berens |
| Deutsche Forschungsgemeinschaft | EXC 2064 - 390727645 | Philipp Berens |
| Deutsche Forschungsgemeinschaft | BE5601/4-1 | Philipp Berens |
| Baden-Württemberg Stiftung | NEU013 | Günther Zeck Philipp Berens |
| National Institutes of Health | EY022070 | Robert G Smith |
| National Institutes of Health | EY023766 | Robert G Smith |

The funders had no role in study design, data collection and interpretation, or the decision to submit the work for publication.

### Author contributions

Jonathan Oesterle, Software, Formal analysis, Investigation, Visualization, Methodology, Writing - original draft, Writing - review and editing; Christian Behrens, Supervision, Investigation, Writing - review and editing; Cornelius Schröder, Methodology, Writing - review and editing; Thoralf Hermann, Katrin Franke, Data curation, Investigation; Thomas Euler, Supervision, Writing - review and editing; Robert G Smith, Conceptualization, Resources, Funding acquisition, Investigation, Writing - review and editing; Günther Zeck, Conceptualization, Supervision, Funding acquisition, Writing - review and editing; Philipp Berens, Conceptualization, Supervision, Funding acquisition, Writing - original draft, Writing - review and editing

### Author ORCIDs

Jonathan Oesterle  https://orcid.org/0000-0001-8919-1445
Christian Behrens  https://orcid.org/0000-0003-3623-352X
Thomas Euler  https://orcid.org/0000-0002-4567-6966
Robert G Smith  https://orcid.org/0000-0001-5703-1324
Günther Zeck  https://orcid.org/0000-0003-3998-9883
Philipp Berens  https://orcid.org/0000-0002-0199-4727

### Ethics

Animal experimentation: All procedures were approved by the governmental review board (Regierungspräsidium Tübingen, Baden-Württemberg, Konrad-Adenauer-Str. 20, 72072 Tübingen, Germany, AZ 35/9185.82-7) and performed according to the laws governing animal experimentation issued by the German Government.

### Decision letter and Author response

Decision letter https://doi.org/10.7554/eLife.54997.sa1
Author response https://doi.org/10.7554/eLife.54997.sa2

## Additional files

### Supplementary files
• Transparent reporting form

### Data availability

Models and simulation code is available at https://github.com/berenslab/CBC_inference (copy archived at https://archive.softwareheritage.org/swh:1:rev:2b8ec4ac0ca916d42c-ba0404229298f8ff79c3a3/). Experimental and inference data is available at https://zenodo.org/record/4185955.

The following dataset was generated:

| Author(s) | Year | Dataset title | Dataset URL | Database and Identifier |
|---|---|---|---|---|
| Oesterle J, Behrens C, Schröder C, Herrmann T, Euler T, Franke K, Smith RG, Zeck G, Berens P | 2020 | Data for "Bayesian inference for biophysical neuron models enables stimulus optimization for retinal neuroprosthetics" | https://zenodo.org/record/4185955 | Zenodo, 10.5281/zenodo.4185955 |

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

# Appendix 1

## Cone prior parameters

**Appendix 1—table 1.** Cone model parameter priors.

| Parameter | Unit | $a_i$ | $\mu_i$ | $b_i$ | References: direct and additional |
|---|---|---|---|---|---|
| $C_m$ | $\mu\mathrm{F\,cm}^{-2}$ | 0.9 | 1 | 1.3 | *Oltedal et al., 2009* |
| $R_m$ | $\mathrm{k\Omega\,cm}^2$ | 1 | 5 | 100 | *Oltedal et al., 2009* |
| $V_r$ | mV | -90 | -70 | -50 | |
| $\mathrm{Ca_L}$ | $\mathrm{mS\,cm}^{-2}$ | 0.1 | 2 | 10 | *Morgans et al., 2005*; *Mansergh et al., 2005*; *Cui et al., 2012* |
| $\mathrm{K_V}$ | $\mathrm{mS\,cm}^{-2}$ | 0 | 0.1 | 3 | *Van Hook et al., 2019* |
| $\mathrm{HCN}@S$ | $\mathrm{mS\,cm}^{-2}$ | 0 | 3 | 10 | *Knop et al., 2008*; *Hellmer et al., 2016*; *Van Hook et al., 2019* |
| $\mathrm{HCN}@A$ | $\mathrm{mS\,cm}^{-2}$ | 0 | 1 | 10 | *Knop et al., 2008*; *Hellmer et al., 2016*; *Van Hook et al., 2019* |
| $\mathrm{HCN}@AT$ | $\mathrm{mS\,cm}^{-2}$ | 0 | 0.1 | 10 | *Knop et al., 2008*; *Hellmer et al., 2016*; *Van Hook et al., 2019* |
| $\mathrm{Cl_{Ca}}$ | $\mathrm{mS\,cm}^{-2}$ | 0 | 0.1 | 10 | *Yang et al., 2008*; *Caputo et al., 2015* |
| $\mathrm{Ca_P}$ | $\mu\mathrm{S\,cm}^{-2}$ | 0.1 | 10 | 100 | *Morgans et al., 1998* |
| $\tau_\alpha(\mathrm{Ca_L})$ | | 0.75 | 1 | 1.5 | |
| $\tau_\alpha(\mathrm{K_V})$ | | 0.1 | 1 | 10 | |
| $\Delta V_\alpha(\mathrm{Ca_L})$ | mV | $-5$ | 0 | 5 | |
| $\Delta V_\alpha(\mathrm{K_V})$ | mV | $-10$ | 0 | 10 | |
| $\mathrm{Ca_{PK}}$ | $\mu\mathrm{mol}$ | 0.01 | 5 | 100 | |
| $v_{RRP}^{max}$ | vesicles | 10 | 20 | 30 | *Thoreson et al., 2016*; *Bartoletti et al., 2010* |
| $g_l$ | | 0.3 | 1 | 3 | |

## BC prior parameters

**Appendix 1—table 2.** BC model parameter priors.

| Parameter | Unit | $a_i$ (3a\|5) | $\mu_i$ (3a\|5) | $b_i$ (3a \| 5) | References: direct and additional |
|---|---|---|---|---|---|
| $C_m$ | $\mu\mathrm{F\,cm}^{-2}$ | 0.9 | 1.18 | 1.3 | *Oltedal et al., 2009* |
| $R_m$ | $\mathrm{k\Omega\,cm}^2$ | 1 | 26 | 100 | *Oltedal et al., 2009* |
| $V_r$ | mV | $-90$ | $-70$ | $-50$ | |
| $\mathrm{Ca_L}@S$ | $\mathrm{mS\,cm}^{-2}$ | 0 | 0.5 | 3 | *Cui et al., 2012, Van Hook et al., 2019* |
| $\mathrm{Ca_L}@AT$ | $\mathrm{mS\,cm}^{-2}$ | 0.1 | 0.5 | 3 | *Cui et al., 2012, Van Hook et al., 2019* |
| $\mathrm{Ca_T}@S$ | $\mathrm{mS\,cm}^{-2}$ | 0\|n/a | 0.5\|n/a | 3\|n/a | *Cui et al., 2012, Van Hook et al., 2019*; *Hu et al., 2009*; *Satoh et al., 1998* |
| $\mathrm{Ca_T}@AT$ | $\mathrm{mS\,cm}^{-2}$ | 0\|n/a | 0.5\|n/a | 3\|n/a | *Cui et al., 2012, Van Hook et al., 2019*; *Hu et al., 2009*; *Satoh et al., 1998* |
| $\mathrm{K_V}@D$ | $\mathrm{mS\,cm}^{-2}$ | 0 | 0.4 | 2 | *Ma et al., 2005* |
| $\mathrm{K_V}@PA$ | $\mathrm{mS\,cm}^{-2}$ | 0 | 0.4 | 2 | *Ma et al., 2005* |
| $\mathrm{K_V}@A$ | $\mathrm{mS\,cm}^{-2}$ | 0 | 0.4 | 2 | *Ma et al., 2005* |
| $\mathrm{K_{ir}}@S$ | $\mathrm{mS\,cm}^{-2}$ | 0 | 1 | 2 | *Cui and Pan, 2008, Knop et al., 2008* |
| $\mathrm{HCN}@D$ | $\mathrm{mS\,cm}^{-2}$ | 0 | 0.2 | 2 | *Hellmer et al., 2016, Knop et al., 2008* |
| $\mathrm{HCN}@S$ | $\mathrm{mS\,cm}^{-2}$ | 0 | 0.2 | 2 | *Hellmer et al., 2016, Knop et al., 2008* |

*Continued on next page*

*Appendix 1—table 2 continued*

| Parameter | Unit | $a_i$ (3a\|5) | $\mu_i$ (3a\|5) | $b_i$ (3a \| 5) | References: direct and additional |
|---|---|---|---|---|---|
| HCN @ $AT$ | mS cm$^{-2}$ | 0 | 0.2 | 2 | *Hellmer et al., 2016, Knop et al., 2008* |
| Na$_V$ @ $DA$ | mS cm$^{-2}$ | 0 | 20 | 100 | *Hellmer et al., 2016* |
| Ca$_P$ @ $S$ | $\mu$S cm$^{-2}$ | 0.1 | 10 | 100 | *Morgans et al., 1998* |
| Ca$_P$ @ $AT$ | $\mu$S cm$^{-2}$ | 0.1 | 10 | 100 | *Morgans et al., 1998* |
| $\tau_\gamma$(Kainate) | | 1\|n/a | 5\|n/a | 20\|n/a | |
| $\tau_\alpha$(Ca$_L$) | | 0.5 | 1 | 2 | |
| $\tau_\alpha$(Ca$_T$) | | 0.5\|n/a | 1\|n/a | 2\|n/a | |
| $\tau_\alpha$(K$_V$) | | 0.1 | 1 | 10 | |
| $\tau_{all}$(Na$_V$) | | 0.5 | 1 | 2 | |
| $\Delta V_\alpha$(Ca$_L$) | mV | −10 | 0 | 10 | |
| $\Delta V_\alpha$(Ca$_T$) | mV | −10\|n/a | 0\|n/a | 10\|n/a | |
| $\Delta V_\alpha$(K$_V$) | mV | −10 | 0 | 10 | |
| $\Delta V_\alpha$(K$_{ir}$) | mV | −5 | 0 | 5 | |
| $\Delta V_\alpha$(Na$_V$) | mV | −5 | 0 | 5 | |
| $\Delta V_\gamma$(Na$_V$) | mV | −5 | 0 | 5 | |
| $Ca_{PK}$ | $\mu$mol | 0.1\|0.01 | 20 | 100 | |
| $STC$ | mmol | 0.05\|1 | 0.5\|1.5 | 1\|3 | |
| $v_{RRP}^{max}$ | vesicles | 4 | 8 | 15 | *Singer and Diamond, 2006* |
| $g_l$ | | 0.5 | 1 | 3 | |

## NeuronC parameters

**Appendix 1—table 3.** Other NeuronC parameters.

| Parameter | Unit | Value | Remarks |
|---|---|---|---|
| vna | mV | 65 | Reversal potential sodium |
| vk | mV | −89 | Reversal potential potassium |
| vcl | mV | −70 | Reversal potential chloride |
| dnao | mmol | 151.5 | Extracellular sodium concentration |
| dko | mmol | 2.5 | Extracellular potassium concentration |
| dclo | mmol | 133.5 | Extracellular chloride concentration |
| dcao | mmol | 2 | Extracellular calcium concentration |
| dicafrac | | 1 | Fraction of calcium pump current that is added to total current |
| use_ghki | | 1 | Use Goldman-Hodgkin-Katz equation |
| cone_timec | | 0.2 | Time constant of cone phototransduction |
| cone_loopg | | 0.0 | Gain of calcium feedback loop in cones |
| cone_maxcond | nS | 0.2 | Max. conductance of of outer segment |
| timinc | ms | 0.001 \| 0.01 | Simulation time step (*Figure 9*) |
| ploti | ms | 0.01 \| 1 | Recording time step (*Figure 9* and *Figure 10* \| otherwise) |

*Continued on next page*

| Parameter | Unit | Value | Remarks |
|---|---|---|---|
| stiminc | cms | 0.01 | 0.1 | Synaptic time step (*Figure 9* and *Figure 10* | otherwise) |
| srtimestep | ms | 0.001 | 0.01 | 0.1 | Stimulus update time step (*Figure 9* | *Figure 10* | otherwise) |
| predur | s | 5 | $\geq$10 | Simulation time to reach equilibrium potential (Cone | BCs) |

