## [Decision Letter]

**Acceptance summary:**

In your paper, you apply Bayesian inference to estimate parameters of multi-compartmental models of cone and cone bipolar cells from 2P-glutamate imaging data. You next use the resulting compartmental models to optimize electrical stimulation of the retina for neuroprosthetics.

As a major advantage of the parameter estimation procedure, the result is not just a singular point in parameter space resulting in an optimal fit, but instead a likelihood distribution showing how well each parameter is constrained by the data. I see this as a significant step forward compared to previous model parameter estimation procedures.

**Decision letter after peer review:**

Thank you for submitting your article "Bayesian inference for biophysical neuron models enables stimulus optimization for retinal neuroprosthetics" for consideration by *eLife*. Your article has been reviewed by three peer reviewers, including Alexander Borst as the Reviewing Editor and Reviewer #1, and the evaluation has been overseen by John Huguenard as the Senior Editor. The following individual involved in review of your submission has agreed to reveal their identity: Adrienne L Fairhall (Reviewer #3).

The reviewers have discussed the reviews with one another and the Reviewing Editor has drafted this decision to help you prepare a revised submission.

Summary:

In this paper, Oesterle and colleagues apply Bayesian inference to estimate parameters of multi-compartmental models of cone and cone ON and OFF bipolar cells from light stimulation and 2P-glutamate imaging as well as EM reconstructions. The method employed (Sequential Neural Posterior Estimation) makes use of a mixture density network, which is an approach that sounds very promising for work in this domain. Importantly, the resulting model parameters are not only given as points to result in an optimal fit, but are given as a likelihood distribution showing how well a specific parameter is constrained by the data. I see this as a significant step forward compared to previous model parameter estimation procedures.

The authors then extended the work to model the effects of electrical stimulation in view of building a modeling test bed for electrical retinal prostheses. This required data constraints from electrical stimulation and current recording. Using the model, the authors were able to design stimuli that selectively targeted different bipolar cells. This work stands as a useful general contribution as a method and the authors have done a thorough and careful job in their modeling. The extension to predictions for retinal prostheses demonstrates the power of the approach.

Essential revisions:

1) This is a systematic study, primarily aimed to demonstrate the advantage of Bayesian simulation-based inference (BSI) for estimating the biophysical parameters of compartmental models. While BSI is convincingly demonstrated as a powerful tool for this purpose, it is not clear whether it is superior in any way over the present-day most-popular alternative, the Multiple Objective Optimization (MOO) approach (Drukmann et al., 2007) which is presently used for building data-constrained compartmental modelling for a large variety of neuron types, in hundreds of labs. worldwide, including at the Allen Inst. the Blue Brain projects etc. One key missing aspect in the present study is a systematic comparison (at least in the Discussion) between BSI and MOO methods.

2) Another key missing aspect in this work is the lack of biophysical intuitions emerging from the compartmental models built. Specifically, how does the synaptic input from the cones propagate along the ON and the OFF BP cells' model that we see in Figure 1? We actually do not see any signal (synaptic potential) in this work neither its propagation along the different compartments – from the distal dendrites to soma, to axon. Does voltage attenuate significantly along these BP – compartments or are the modeled cell close to isopotentiality? What is the role of active ion channels during signals propagation in these models? What is the synaptic conductances (between Cone and BP cells) in these models (and what is the justification to use such a complicated model for transmitter release with Ca-dependent pool-release, rather than transient (double exponential?) conductance change as synaptic inputs)? What are the key differences between the ON and OFF compt. models that make them respond differentially to extracellular stimulus? The authors write in the Discussion: "Likely, the different density for some ion channels contributed to the differential response of the two BC types". This is clearly an unsounded claim which needs to be shown and discussed. After all – an important usage of compartmental modelling is to gain insights into the interaction between structure/membrane channels and synaptic/input-output properties of the modeled cells and for calibrating the model against experiments. This key aspect is missing in this work and, therefore, it is impossible for the reader to grasp the underlying mechanisms responsible for the emergent properties of the modeled cells in response to light stimulus.

3) Another query is whether the response of the modeled ON and OFF BP cells will not be very different when they are immersed in actual retina circuit, with electrical field generated also by other cell types (AC, GC) when the retina is electrically-stimulated. This point should be discussed.

4) It is essential to provide the readers with Neuron models of the 3-cell types as well as with the respective data that was used to constrain these models

5) While the paper is generally well written and clear, the model exposition (Section entitled "Inference algorithm") leaves considerable room for improvement; ideally the paper should be self-contained. The presentation is a little confusing with respect to the status of *p*(*θ*), the "proposal prior" *p* ~, the posterior *p*(*θ*|x) and the "auxiliary" distribution q (which when equated to the posterior is no longer written as a conditional distribution as it appears in the cost function). It would be good to explain the form of the cost function that is minimized-it looks like it is based on a KL divergence but is a bit unclear of what. This exposition could do a much better job of walking pedagogically through the goal of the algorithm and how the goal is achieved by the variables defined and the cost function. Also, one should shorten this part of the paper and shift many of the figures to Appendices – as it is now standing with 10 figures.

6) It is difficult to find a quantitative reporting of the variation between data from the same cell type. I took the method to be applied to fit distributions over parameters for models accounting for each experimental trace separately; and for the (beautiful!) results in Figures 5 and 6 to be from one example cell, but maybe this is not true. Could this be clarified? Are the distributions of 7 over models that fit all the experimental data for that cell type? If not, it would be good to show the measured responses with an error bar, and show variations between models. Understanding the extent of intercellular variability seems important in the design of isolating stimuli.

---

## [Author Response]

Essential revisions:1) This is a systematic study, primarily aimed to demonstrate the advantage of Bayesian simulation-based inference (BSI) for estimating the biophysical parameters of compartmental models. While BSI is convincingly demonstrated as a powerful tool for this purpose, it is not clear whether it is superior in any way over the present-day most-popular alternative, the Multiple Objective Optimization (MOO) approach (Drukmann et al., 2007) which is presently used for building data-constrained compartmental modelling for a large variety of neuron types, in hundreds of labs. worldwide, including at the Allen Inst. the Blue Brain projects etc. One key missing aspect in the present study is a systematic comparison (at least in the Discussion) between BSI and MOO methods.

We thank the reviewers for giving us the chance to clarify this point. Our contribution in this paper is to show that Bayesian inference for biophysical multicompartment models of neurons is feasible using fairly standard light stimuli and recordings and one can obtain meaningful posterior distributions even for such high-dimensional inference problems.

Indeed, many other labs use MOO algorithms for similar models. However, recent parallel work by Lueckmann et al. also under review at *eLife*

(https://www.biorxiv.org/content/10.1101/838383v3.full.pdf) has shown that already for single compartment Hodgkin-Huxley models MOO algorithms do not in general perform proper statistical inference – in particular, the best simulations obtained by such algorithms are often more concentrated in the parameter space and one cannot easily get a good sense of the uncertainty in the parameters from such algorithms. We expanded our discussion of alternative approaches, highlighting also the comparative analysis in the abovementioned paper. Also, we expanded our analysis (e.g. in Figures 9 and 10) to explicitly take the posterior into account.

2) Another key missing aspect in this work is the lack of biophysical intuitions emerging from the compartmental models built. Specifically, how does the synaptic input from the cones propagate along the ON and the OFF BP cells' model that we see in Figure 1? We actually do not see any signal (synaptic potential) in this work neither its propagation along the different compartments – from the distal dendrites to soma, to axon. Does voltage attenuate significantly along these BP – compartments or are the modeled cell close to isopotentiality? What is the role of active ion channels during signals propagation in these models?

To address this point, we created animated heatmaps overlaid on the BC’s morphology to show the voltage, calcium and glutamate signals as a function of space and time, available as video files, for the chirp stimulus. Snapshots from these videos at discrete time points are available as Figure 6—figure supplements 1 and 2. We discuss this analysis in the Results section. It shows that BCs are relatively isopotential units. We checked for the importance of active ion channels by removing them from the optimized model (calcium channels in the axon terminals were left in, as they are required for simulating release and do not contribute strongly to the voltage). We found that for the cone model, the calcium-activated chloride channels did almost not contribute to the light response in the optimized model, neither for the cone nor the BC stimulus. We therefore removed it in the following steps, to save computational resources. For the BCs, we extended this analysis to the five best posterior samples (and not only the best sample). We found that, in both cells, the sodium channel (Na_V_) and somatic calcium channels could be removed from the model without altering the light response. Separate from the calcium channels in the axon terminals, the HCN was most significant for shaping the light response. We discuss all these findings in the paper now.

What is the synaptic conductances (between Cone and BP cells) in these models?

This parameter was missing from the manuscript. We now provide it in the Materials and methods section.

What is the justification to use such a complicated model for transmitter release with Ca-dependent pool-release, rather than transient (double exponential?) conductance change as synaptic inputs?

Ribbon synapses are a hallmark of glutamate release in cones and bipolar cells (for review, see Baden et al., 2011, https://doi.org/10.1016/j.tins.2013.04.006). The dynamics of release at this type of synapse cannot be easily matched with a simple transient conductance change, as the state of the pools determines the kinetics of the release. For example, if all pools are filled, ribbon synapses can release a large amount of glutamate very rapidly, emphasizing transient signals. We added a sentence justifying this choice in the Materials and methods section.

What are the key differences between the ON and OFF compt. models that make them respond differentially to extracellular stimulus? The authors write in the Discussion: "Likely, the different density for some ion channels contributed to the differential response of the two BC types". This is clearly an unsounded claim which needs to be shown and discussed.

We investigated this finding in more detail by varying additionally the cell position relative to the electrode and sampling cell parameters from the posterior. We found that the optimal stimulus waveform for the OFF cell was robust against changes in the cell position relative to the electrode and the precise choice model parameters, while the ON cell could less reliably be activated exclusively, with typical stimuli good for ON cells also evoking sizable OFF cell responses (similar to what could be seen in the original Figure 10). This different behavior was due to the T-type calcium channel in the OFF-BC, which strongly reacts to transient stimulation. The ON-BC can only be selectively activated if the threshold for the OFF-BC is larger, which depends on the cell position and the exact cell parametrization. These new results are now discussed in the paper.

After all – an important usage of compartmental modelling is to gain insights into the interaction between structure/membrane channels and synaptic/input-output properties of the modeled cells and for calibrating the model against experiments. This key aspect is missing in this work and, therefore, it is impossible for the reader to grasp the underlying mechanisms responsible for the emergent properties of the modeled cells in response to light stimulus.

As detailed above, we added several new figures and performed new analysis following the feedback of the reviewer. We feel that these have improved the paper significantly.

3) Another query is whether the response of the modeled ON and OFF BP cells will not be very different when they are immersed in actual retina circuit, with electrical field generated also by other cell types (AC, GC) when the retina is electrically-stimulated. This point should be discussed.

The reviewers are right that being able to selectively stimulate OFF/ON BCs in isolation is not the same as being able to stimulate them selectively when embedded in the retinal network. Of course, network effects through synaptic activation of ACs will further modulate the activity evoked by the stimulation. It is unlikely that ACs or RGCs get directly activated by the electrical stimulation protocol suggested here, as their neurites and somata are substantially farther away from the stimulation electrode than those of BCs, and Figures 3D and 10 illustrate that even for BCs mostly the dendrites are directly affected by the electric field. That notwithstanding, simulations including network effects and also more diverse BC types will be required in the future. We added this to the Discussion.

4) It is essential to provide the readers with Neuron models of the 3-cell types as well as with the respective data that was used to constrain these models

We will make the code and the data available upon publication at our github account https://github.com/berenslab and will provide the version used to create the figures for the paper as version 1.0 archived on zenodo.

5) While the paper is generally well written and clear, the model exposition (Section entitled "Inference algorithm") leaves considerable room for improvement; ideally the paper should be self-contained. The presentation is a little confusing with respect to the status of p(θ), the "proposal prior" p ~, the posterior p(θ|x) and the "auxiliary" distribution q (which when equated to the posterior is no longer written as a conditional distribution as it appears in the cost function). It would be good to explain the form of the cost function that is minimized-it looks like it is based on a KL divergence but is a bit unclear of what. This exposition could do a much better job of walking pedagogically through the goal of the algorithm and how the goal is achieved by the variables defined and the cost function. Also, one should shorten this part of the paper and shift many of the figures to Appendices – as it is now standing with 10 figures.

We reordered this part of the Materials and methods section. Now the section starts with the target data and we explain the discrepancy measure between simulations and target data. We then explained the inference algorithm step-by-step and changed the wording to make it clearer. We feel that moving figures from the Materials and methods section to the Appendix would rather make the paper harder to read.

6) It is difficult to find a quantitative reporting of the variation between data from the same cell type. I took the method to be applied to fit distributions over parameters for models accounting for each experimental trace separately; and for the (beautiful!) results in Figures 5 and 6 to be from one example cell, but maybe this is not true. Could this be clarified? Are the distributions of 7 over models that fit all the experimental data for that cell type? If not, it would be good to show the measured responses with an error bar, and show variations between models. Understanding the extent of intercellular variability seems important in the design of isolating stimuli.

Indeed, the models were fit to the BC type means, as individual recordings can be quite noisy – we added Figure 5—figure supplement 1 to illustrate the variability and mention this in the text.